# Provably Faster Algorithms for Bilevel Optimization

**Junjie Yang**
Department of ECE
The Ohio State University
`yang.4972@osu.edu`

**Kaiyi Ji**
Department of EECS
University of Michigan
`kaiyiji@umich.edu`

**Yingbin Liang**
Department of ECE
The Ohio State University
`liang.889@osu.edu`

## Abstract

Bilevel optimization has been widely applied in many important machine learning applications such as hyperparameter optimization and meta-learning. Recently, several momentum-based algorithms have been proposed to solve bilevel optimization problems faster. However, those momentum-based algorithms do not achieve provably better computational complexity than $\widetilde{\mathcal{O}}(\epsilon^{-2})$ of the SGD-based algorithm. In this paper, we propose two new algorithms for bilevel optimization, where the first algorithm adopts momentum-based recursive iterations, and the second algorithm adopts recursive gradient estimations in nested loops to decrease the variance. We show that both algorithms achieve the complexity of $\widetilde{\mathcal{O}}(\epsilon^{-1.5})$, which outperforms all existing algorithms by the order of magnitude. Our experiments validate our theoretical results and demonstrate the superior empirical performance of our algorithms in hyperparameter applications.

## 1 Introduction

Bilevel optimization has become a timely and important topic recently due to its great effectiveness in a wide range of applications including hyperparameter optimization [7, 5], meta-learning [33, 16, 1], reinforcement learning [14, 24]. Bilevel optimization can be generally formulated as the following minimization problem:

$$\min_{x\in\mathbb{R}^p}\Phi(x) := f(x,y^*(x)) \quad \text{s.t.} \ \ y^*(x) = \arg\min_{y\in\mathbb{R}^q} g(x,y). \tag{1}$$

Since the outer function $\Phi(x) := f(x,y^*(x))$ depends on the variable $x$ also via the optimizer $y^*(x)$ of the inner-loop function $g(x,y)$, the algorithm design for bilevel optimization is much more complicated and challenging than minimization and minimax optimization. For example, if the gradient-based approach is applied, then the gradient of the outer-loop function (also called *hypergradient*) will necessarily involve Jacobian and Hessian matrices of the inner-loop function $g(x,y)$, which require more careful design to avoid high computational complexity.

This paper focuses on the nonconvex-strongly-convex setting, where the outer function $f(x,y^*(x))$ is nonconvex with respect to (w.r.t.) $x$ and the inner function $g(x,y)$ is strongly convex w.r.t. $y$ for any $x$. Such a case often occurs in practical applications. For example, in hyperparameter optimization [7], $f(x,y^*(x))$ is often nonconvex with $x$ representing neural network hyperparameters, but the inner function $g(x,\cdot)$ can be strongly convex w.r.t. $y$ by including a strongly-convex regularizer on $y$. In few-shot meta-learning [1], the inner function $g(x,\cdot)$ often takes a quadratic form together with a strongly-convex regularizer. To efficiently solve the deterministic problem in eq. (1), various bilevel optimization algorithms have been proposed, which include two popular classes of deterministic gradient-based methods respectively based on approximate implicit differentiation (AID) [31, 9, 8] and iterative differentiation (ITD) [28, 6, 7].

35th Conference on Neural Information Processing Systems (NeurIPS 2021).

Recently, stochastic bilevel opitimizers [8, 20] have been proposed, in order to achieve better efficiency than deterministic methods for large-scale scenarios where the data size is large or vast fresh data needs to be sampled as the algorithm runs.

In particular, such a class of problems adopt functions by:

$$\Phi(x) := f(x, y^*(x)) := \mathbb{E}_\xi[F(x, y^*(x); \xi)], \quad g(x, y) := \mathbb{E}_\zeta[G(x, y; \zeta)]$$

where the outer and inner functions take the expected values w.r.t. samples $\xi$ and $\zeta$, respectively.

Along this direction, [20] proposed a stochastic gradient descent (SGD) type optimizer (stocBiO), and showed that stocBiO attains a computational complexity of $\widetilde{\mathcal{O}}(\epsilon^{-2})$ in order to reach an $\epsilon$-accurate stationary point. More recently, several studies [2, 11, 22] have tried to accelerate SGD-type bilevel optimizers via momentum-based techniques, e.g., by introducing a momentum (historical information) term into the gradient estimation. All of these optimizers follow a **single-loop** design, i.e., updating $x$ and $y$ simultaneously. Specifically, [22] proposed an algorithm MSTSA by updating $x$ via a momentum-based recursive technique introduced by [3, 36]. [11] proposed an optimizer SEMA similarly to MSTSA but using the momentum recursive technique for updating both $x$ and $y$. [2] proposed an algorithm STABLE, which applies the momentum strategy for updating the Hessian matrix, but the algorithm involves expensive Hessian inverse computation rather than hypergradient approximation loop. However, as shown in Table 1, SEMA, MSTSA and STABLE achieve the same complexity order of $\widetilde{\mathcal{O}}(\epsilon^{-2})$ as the SGD-type stocBiO algorithm, where the momentum technique in these algorithms does not exhibit the theoretical advantage. Such a comparison is not consistent with those in minimization [3] and minimax optimization [15], where the single-loop momentum-based recursive technique achieves provable performance improvements over SGD-type methods. This motivates the following natural but important question:

- Can we design a faster single-loop momentum-based recursive bilevel optimizer, which achieves order-wisely lower computational complexity than SGD-type stocBiO (and all other momentum-based algorithms), and is also easy to implement with efficient matrix-vector products?

Although the existing theoretical efforts on accelerating bilevel optimization algorithms have been exclusively focused on single-loop design[1], empirical studies in [20] suggested that **double-loop** bilevel algorithms such as BSA [8] and stocBiO [20] achieve much better performances than **single-loop** algorithms such as TTSA [14]. A good candidate suitable for accelerating double-loop algorithms can be the popular variance reduction method, such as SVRG [21], SARAH [30] and SPIDER [4], which typically yield provably lower complexity. The basic idea is to construct low-variance gradient estimators using periodic high-accurate large-batch gradient evaluations. So far, there has not been any study on using variance reduction to accelerate double-loop bilevel optimization algorithms. This motivates the second question that we address in this paper:

- Can we develop a double-loop variance-reduced bilevel optimizer with improved computational complexity over SGD-type stocBiO (and all other existing algorithms)? If so, when such a **double-loop** algorithm holds advantage over the **single-loop** algorithms in bilevel optimization?

## 1.1 Main Contributions

This paper proposes two algorithms for bilevel optimization, both outperforming all existing algorithms in terms of complexity order.

We first propose a single-loop momentum-based recursive bilevel optimizer (MRBO). MRBO updates variables $x$ and $y$ simultaneously, and uses the momentum recursive technique for constructing low-variance **mini-batch** estimators for both the gradient $\nabla g(x, \cdot)$ and the hypergradient $\nabla \Phi(\cdot)$; in contrast to previous momentum-based algorithms that accelerate only one gradient or neither. Further, MRBO is easy to implement, and allows efficient computations of Jacobian- and Hessian-vector products via automatic differentiation. Theoretically, we show that MRBO achieves a computational complexity (w.r.t. computations of gradient, Jacobian- and Hessian-vector product) of $\widetilde{\mathcal{O}}(\epsilon^{-1.5})$,

---

[1]In the literature of bilevel optimization, although many hypergradient-based algorithms include an iteration loop of Hessian inverse estimation, such a loop is typically not counted when these algorithms are classified by the number of loops. This paper follows such a convention to be consistent with the existing literature. Namely, the single- and double-loop algorithms mentioned here can include an additional loop of Hessian inverse estimation in the hypergradient approximation.

Table 1: Comparison of stochastic algorithms for bilevel optimization.

| Algorithm | $Gc(F, \epsilon)$ | $Gc(G, \epsilon)$ | $JV(G, \epsilon)$ | $HV(G, \epsilon)$ | $Hyy^{inv}(G, \epsilon)$ |
|---|---|---|---|---|---|
| MSTSA [22] | $\mathcal{O}(\epsilon^{-2})$ | $\mathcal{O}(\epsilon^{-2})$ | $\mathcal{O}(\epsilon^{-2})$ | $\widetilde{\mathcal{O}}(\epsilon^{-2})$ | / |
| SEMA [11] | $\widetilde{\mathcal{O}}(\epsilon^{-2})$ | $\widetilde{\mathcal{O}}(\epsilon^{-2})$ | $\widetilde{\mathcal{O}}(\epsilon^{-2})$ | $\widetilde{\mathcal{O}}(\epsilon^{-2})$ | / |
| STABLE [2] | $\mathcal{O}(\epsilon^{-2})$ | $\mathcal{O}(\epsilon^{-2})$ | / | / | $\mathcal{O}(\epsilon^{-2})$ |
| stocBiO [20] | $\mathcal{O}(\epsilon^{-2})$ | $\mathcal{O}(\epsilon^{-2})$ | $\mathcal{O}\left(\epsilon^{-2}\right)$ | $\widetilde{\mathcal{O}}\left(\epsilon^{-2}\right)$ | / |
| RSVRB [12] (Concurrent) | $\mathcal{O}(\epsilon^{-1.5})$ | $\mathcal{O}(\epsilon^{-1.5})$ | $\mathcal{O}\left(\epsilon^{-1.5}\right)$ | / | $\mathcal{O}\left(\epsilon^{-1.5}\right)$ |
| SUSTAIN [23] (Concurrent) | $\mathcal{O}(\epsilon^{-1.5})$ | $\mathcal{O}(\epsilon^{-1.5})$ | $\mathcal{O}\left(\epsilon^{-1.5}\right)$ | $\widetilde{\mathcal{O}}\left(\epsilon^{-1.5}\right)$ | / |
| MRBO (ours) | $\mathcal{O}(\epsilon^{-1.5})$ | $\mathcal{O}(\epsilon^{-1.5})$ | $\mathcal{O}\left(\epsilon^{-1.5}\right)$ | $\widetilde{\mathcal{O}}\left(\epsilon^{-1.5}\right)$ | / |
| VRBO (ours) | $\widetilde{\mathcal{O}}(\epsilon^{-1.5})$ | $\widetilde{\mathcal{O}}(\epsilon^{-1.5})$ | $\widetilde{\mathcal{O}}\left(\epsilon^{-1.5}\right)$ | $\widetilde{\mathcal{O}}\left(\epsilon^{-1.5}\right)$ | / |

$Gc(F, \epsilon)$ and $Gc(G, \epsilon)$: number of gradient evaluations w.r.t. $F$ and $G$.

$Jv(G, \epsilon)$: number of Jacobian-vector products $\nabla_x \nabla_y G(\cdot)v$. $\widetilde{\mathcal{O}}(\cdot)$: omit $\log \frac{1}{\epsilon}$ terms.

$Hv(G, \epsilon)$: number of Hessian-vector products $\nabla_y^2 G(\cdot)v$.

$Hyy^{inv}(G, \epsilon)$: number of evaluations of Hessian inverse $[\nabla_y^2 G]^{-1}$.

which outperforms all existing algorithms by an order of $\epsilon^{-0.5}$. Technically, our analysis needs to first characterize the estimation property for the momentum-based recursive estimator for the **Hessian-vector type** hypergradient and then uses such a property to further bound the per-iteration error due to momentum updates for both inner and outer loops.

We then propose a double-loop variance-reduced bilevel optimizer (VRBO), which is the first algorithm that adopts the recursive variance reduction for bilevel optimization. In VRBO, each inner loop constructs a variance-reduced gradient (w.r.t. $y$) and **hypergradient** (w.r.t. $x$) estimators through the use of large-batch gradient estimations computed periodically at each outer loop. Similarly to MRBO, VRBO involves the computations of Jacobian- and Hessian-vector products rather than Hessians or Hessian inverse. Theoretically, we show that VRBO achieves the same near-optimal complexity of $\widetilde{\mathcal{O}}(\epsilon^{-1.5})$ as MRBO and outperforms all existing algorithms. Technically, differently from the use of variance reduction in minimization and minimax optimization, our analysis for VRBO needs to characterize the variance reduction property for the **Hessian-vector type** of hypergradient estimators, which only involves Hessian vector computation rather than Hessian. Such estimator introduces additional errors to handle in the telescoping and convergence analysis.

Our experiments[2] show that VRBO achieves the highest accuracy among all comparison algorithms, and MRBO converges fastest among its same type of single-loop momentum-based algorithms. In particular, we find that our double-loop VRBO algorithm converges much faster than other singlr-loop algorithms including our MRBO, which is in contrast to the existing efforts exclusively on accelerating the single-loop algorithms [2, 11, 22]. Such a result also differs from those phenomenons observed in minimization and minimax optimization, where single-loop algorithms often outperform double-loop algorithms.

## 1.2 Related Works

**Bilevel optimization approaches:** At the early stage of bilevel optimization studies, a class of constraint-based algorithms [13, 35, 29] were proposed, which tried to penalize the outer function with the optimality conditions of the inner problem. To further simplify the implementation of constraint-based bilevel methods, gradient-based bilevel algorithms were then proposed, which include but not limited to AID-based [33, 7, 34, 17], ITD-based [9, 31, 8, 16, 19] methods, and stochastic bilevel optimizers such as BSA [8], stocBiO [20], and TTSA [14]. The finite-time (i.e., non-asymptotic) convergence analysis for bilevel optimization has been recently studied in several works [8, 20, 14]. In this paper, we propose two novel stochastic bilevel algorithms using momentum recursive and variance reduction techniques, and show that they order-wise improve the computational complexity over existing stochastic bilevel optimizers.

**Momentum-based recursive approaches:** The momentum recursive technique was first introduced by [3, 36] for minimization problems, and has been shown to achieve improved computational complexity over SGD-based updates in theory and in practice. Several works [22, 2, 11] applied the

---

[2]Our codes are available online at https://github.com/JunjieYang97/MRVRBO

similar single-loop momentum-based strategy to bilevel optimization to accelerate the SGD-based bilevel algorithms such as BSA [8] and stocBiO [20]. However, the computational complexities of these momentum-based algorithms are not shown to outperform that of stocBiO. In this paper, we propose a new single-loop momentum-based recursive bilevel optimizer (MRBO), which we show achieves order-wisely lower complexity than existing stochastic bilevel optimizers.

**Variance reduction approaches:** Variance reduction has been studied extensively for conventional minimization problems, and many algorithms have been designed along this line, including but not limited to SVRG [21, 26], SARAH [30], SPIDER [4], SpiderBoost [37, 38, 18] and SNVRG [41]. Several works [27, 39, 40, 32] recently employed such techniques for minimax optimization to achieve better complexities. In this paper, we propose the first-known variance reduction-based bilevel optimizer (VRBO), which achieves a near-optimal computational complexity and outperforms existing stochastic bilevel algorithms.

**Two concurrent works:** As we were finalizing this submission, two concurrent studies were posted on arXiv recently ([23] was posted on May 8 and [12] was posted on May 5). Both studies overlap only with our MRBO algorithm, nothing similar to our VRBO. Specifically, [23] and [12] respectively proposed the SUSTAIN and RSVRB algorithms for bilevel optimization, both using momentum-based design as our MRBO. Although SUSTAIN and RSVRB have been shown to achieve the same theoretical complexity of $\mathcal{O}(\epsilon^{-1.5})$ as our MRBO (and VRBO), both algorithms have major drawbacks in their design, so that their empirical performance (as we demonstrate in our experiments) is much worse that our MRBO (and even worse than our VRBO). SUSTAIN adopts only single-sample for each update (whereas MRBO uses minibatch for stability); and RSVRB requires to compute Hessian inverse at each iteration (whereas MRBO uses Hessian-vector products for fast computation). As an additional note, our experiments demonstrate that our VRBO significantly outperforms all these single-loop momentum-based algorithms SUSTAIN and RSVRB as well as our MRBO.

## 2 Two New Algorithms

In this section, we propose two new algorithms for bilevel optimization. Firstly, we introduce the hypergradient of the objective function $\Phi(x_k)$, which is useful for designing stochastic algorithms.

**Property 1.** *The (hyper)gradient of $\Phi(x) = f(x, y^*(x))$ in eq. (1) takes a form of*

$$\nabla\Phi(x) = \nabla_x f(x, y^*(x)) - \nabla_x \nabla_y g(x, y^*(x))[\nabla_y^2 g(x, y^*(x))]^{-1} \nabla_y f(x, y^*(x)). \quad (2)$$

However, it is not necessary to compute $y^*$ for updating $x$ at every iteration, and it is not time and memory efficient to compute Hessian inverse matrix in eq. (2) explicitly. Here, we estimate the hypergradient similarly to [20, 8], which takes a form of

$$\overline{\nabla}\Phi(x) = \nabla_x f(x, y) - \nabla_x \nabla_y g(x, y)\eta \sum_{q=-1}^{Q-1} (I - \eta\nabla_y^2 g(x, y))^{q+1} \nabla_y f(x, y), \quad (3)$$

where the Neumann series $\eta \sum_{i=0}^{\infty}(I - \eta G)^i = G^{-1}$ is applied to approximate the Hessian inverse.

### 2.1 Momentum-based Recursive Bilevel Optimizer (MRBO)

As shown in Algorithm 1, we propose a **M**omentum-based **R**ecursive **B**ilevel **O**ptimizer (MRBO) for solving the bilevel problem in eq. (1).

MRBO updates in a single-loop manner, where the momentum recursive technique STORM [3] is employed for updating both $x$ and $y$ at each iteration simultaneously. To update $y$, at step $k$, MRBO first constructs the momentum-based gradient estimator $u_k$ based on the current $\nabla_y G(x_k, y_k; \mathcal{B}_y)$ and the previous $\nabla_y G(x_{k-1}, y_{k-1}; \mathcal{B}_y)$ using a minibatch $\mathcal{B}_y$ of samples (see line 8 in Algorithm 1). Note that the hyperparameter $\beta_k$ decreases at each iteration, so that the gradient estimator $u_k$ is more determined by the previous $u_{k-1}$, which improves the stability of gradient estimation, especially when $y_k$ is close to the optimal point. Then MRBO uses the gradient estimator for updating $y_k$ (see line 11). The stepsize $\eta_k$ decreases at each iteration to reduce the convergence error.

To update $x$, at step $k$, MRBO first constructs the momentum-based recursive hypergradient estimator $v_k$ based on the current $\widehat{\nabla}\Phi(x_k; \mathcal{B}_x)$ and the previous $\widehat{\nabla}\Phi(x_{k-1}; \mathcal{B}_x)$ computed using several

---

**Algorithm 1** Momentum-based Recursive Bilevel Optimizer (MRBO)

---
1: **Input:** Stepsize $\lambda, \gamma > 0$, Coefficients $\alpha_0, \beta_0$, Initializers $x_0, y_0$, Hessian Estimation Number $Q$, Batch Size $S$, Constant $c_1, c_2, m, d > 0$
2: **for** $k = 0, 1, \ldots, K$ **do**
3:     Draw Samples $\mathcal{B}_y, \mathcal{B}_x = \{\mathcal{B}_j (j = 1, \ldots, Q), \mathcal{B}_F, \mathcal{B}_G\}$ with batch size $S$ for each component
4:     **if** $k = 0$: **then**
5:         $v_k = \widehat{\nabla}\Phi(x_k; \mathcal{B}_x), u_k = \nabla_y G(x_k, y_k; \mathcal{B}_y)$
6:     **else**
7:         $v_k = \widehat{\nabla}\Phi(x_k; \mathcal{B}_x) + (1 - \alpha_k)(v_{k-1} - \widehat{\nabla}\Phi(x_{k-1}; \mathcal{B}_x))$
8:         $u_k = \nabla_y G(x_k, y_k; \mathcal{B}_y) + (1 - \beta_k)(u_{k-1} - \nabla_y G(x_{k-1}, y_{k-1}; \mathcal{B}_y))$
9:     **end if**
10:    **update:** $\eta_k = \frac{d}{\sqrt[3]{m+k}}, \quad \alpha_{k+1} = c_1 \eta_k^2, \quad \beta_{k+1} = c_2 \eta_k^2$
11:    $x_{k+1} = x_k - \gamma \eta_k v_k, \quad y_{k+1} = y_k - \lambda \eta_k u_k$
12: **end for**

---

independent minibatches of samples $\mathcal{B}_x = \{\mathcal{B}_j (j = 1, \ldots, Q), \mathcal{B}_F, \mathcal{B}_G\}$ (see line 7 in Algorithm 1). The hyperparameter $\alpha_k$ decreases at each iteration, so that the new gradient estimation $v_k$ is more determined by the previous $v_{k-1}$, which improves the stability of gradient estimation, especially when $x_k$ is around the optimal point. Specifically, the hypergradient estimator $\widehat{\nabla}\Phi(x_k; \mathcal{B}_x)$ is designed based on the expected form in eq. (3), and takes a form of:

$$\widehat{\nabla}\Phi(x_k; \mathcal{B}_x) = \nabla_x F(x_k, y_k; \mathcal{B}_F)$$
$$- \nabla_x \nabla_y G(x_k, y_k; \mathcal{B}_G) \eta \sum_{q=-1}^{Q-1} \prod_{j=Q-q}^{Q} (I - \eta \nabla_y^2 G(x_k, y_k; \mathcal{B}_j)) \nabla_y F(x_k, y_k; \mathcal{B}_F), \quad (4)$$

Note that MRBO computes the above estimator recursively using only **Hessian vectors** rather than **Hessians** (see Appendix A) in order to reduce the memory and computational cost. Then MRBO uses the estimated gradient $v_k$ for updating $x_k$ (see line 11). The stepsize $\eta_k$ decreases at each iteration to facilitate the convergence.

## 2.2 Variance Reduction Bilevel Optimizer (VRBO)

Although all of the existing momentum algorithms [2, 22, 11] (and two current studies [23, 12]) for bilevel optimization follow the single-loop design, empirical results in [20] suggest that **double-loop** bilevel algorithms can achieve much better performances than **single-loop** algorithms. Thus, as shown in Algorithm 2, we propose a double-loop algorithm called **V**ariance **R**eduction **B**ilevel **O**ptimizer (VRBO). VRBO adopts the variance reduction technique in SARAH [30]/SPIDER [4] for bilevel optimization, which is suitable for designing double-loop algorithms. Specifically, VRBO constructs the recursive variance-reduced gradient estimators for updating both $x$ and $y$, where each update of $x$ in the outer-loop is followed by $(m + 1)$ inner-loop updates of $y$. VRBO divides the outer-loop iterations into epochs, and at the beginning of each epoch computes the hypergradient estimator $\widehat{\nabla}\Phi(x_k, y_k; \mathcal{S}_1)$ and the gradient $\nabla_y G(x_k, y_k; \mathcal{S}_1)$ based on a relatively large batch $\mathcal{S}_1$ of samples for variance reduction, where $\widehat{\nabla}\Phi(x_k, y_k; \mathcal{S}_1)$ takes a form of

$$\widehat{\nabla}\Phi(x_k, y_k; \mathcal{S}_1) = \frac{1}{S_1} \sum_{i=1}^{S_1} \Big( \nabla_x F(x_k, y_k; \xi_i)$$
$$- \nabla_x \nabla_y G(x_k, y_k; \zeta_i) \eta \sum_{q=-1}^{Q-1} \prod_{j=Q-q}^{Q} (I - \eta \nabla_y^2 G(x_k, y_k; \zeta_i^j)) \nabla_y F(x_k, y_k; \xi_i) \Big), \quad (5)$$

where all samples in $\mathcal{S}_1 = \{\zeta_i^j (j = 1, \ldots, Q), \xi_i, \zeta_i, i = 1, \ldots, S_1\}$ are independent. Note that eq. (5) takes a different form from MRBO in eq. (4), but the Hessian-vector computation method for MRBO is still applicable here. Then, VRBO recursively updates the gradient estimators for $\nabla_y G(\widetilde{x}_{k,t}, \widetilde{y}_{k,t}; \mathcal{S}_2)$ and $\widehat{\nabla}\Phi(\widetilde{x}_{k,t}, \widetilde{y}_{k,t}; \mathcal{S}_2)$ (which takes the same form as eq. (5)) with a small sample batch $\mathcal{S}_2$ (see lines 11 to 16) during inner-loop iterations.

We remark that VRBO is the first algorithm that adopts the recursive variance reduction method for bilevel optimization. As we will shown in Section 3, VRBO achieves the same nearly-optimal computational complexity as MRBO (and outperforms all other existing algorithms). More interestingly, as a double-loop algorithm, VRBO empirically significantly outperforms all existing single-loop momentum algorithms including MRBO. More details and explanation are provided in Section 4.

---

**Algorithm 2** Variance Reduction Bilevel Optimizer (VRBO)

---

1: **Input:** Stepsize $\beta, \alpha > 0$, Initializer $x_0, y_0$, Hessian $Q$, Sample Size $S_1, S_2$, Periods $q$
2: **for** $k = 0, 1, \ldots, K$ **do**
3:    **if** $\mathrm{mod}(k, q) = 0$: **then**
4:      Draw a batch $\mathcal{S}_1$ of i.i.d. samples
5:      $u_k = \nabla_y G(x_k, y_k; \mathcal{S}_1), v_k = \widehat{\nabla}\Phi(x_k, y_k; \mathcal{S}_1)$
6:    **else**
7:      $u_k = \widetilde{u}_{k-1,m+1}, v_k = \widetilde{v}_{k-1,m+1}$
8:    **end if**
9:    $x_{k+1} = x_k - \alpha v_k$
10:    Set $\widetilde{x}_{k,-1} = x_k, \widetilde{y}_{k,-1} = y_k, \widetilde{x}_{k,0} = x_{k+1}, \widetilde{y}_{k,0} = y_k, \widetilde{v}_{k,-1} = v_k, \widetilde{u}_{k,-1} = u_k$
11:    **for** $t = 0, 1, \ldots, m+1$ **do**
12:      Draw a batch $\mathcal{S}_2$ of i.i.d samples
13:      $\widetilde{v}_{k,t} = \widetilde{v}_{k,t-1} + \widehat{\nabla}\Phi(\widetilde{x}_{k,t}, \widetilde{y}_{k,t}; \mathcal{S}_2) - \widehat{\nabla}\Phi(\widetilde{x}_{k,t-1}, \widetilde{y}_{k,t-1}; \mathcal{S}_2)$
14:      $\widetilde{u}_{k,t} = \widetilde{u}_{k,t-1} + \nabla_y G(\widetilde{x}_{k,t}, \widetilde{y}_{k,t}; \mathcal{S}_2) - \nabla_y G(\widetilde{x}_{k,t-1}, \widetilde{y}_{k,t-1}; \mathcal{S}_2)$
15:      $\widetilde{x}_{k,t+1} = \widetilde{x}_{k,t}, \widetilde{y}_{k,t+1} = \widetilde{y}_{k,t} - \beta \widetilde{u}_{k,t}$
16:    **end for**
17:    $y_{k+1} = \widetilde{y}_{k,m+1}$
18: **end for**

---

## 3 Main Results

In this section, we first introduce several standard assumptions for the analysis, and then present the convergence results for the proposed MRBO and VRBO algorithms.

### 3.1 Technical Assumptions and Definitions

**Assumption 1.** *Assume that the inner function $G(x, y; \zeta)$ is $\mu$-strongly-convex w.r.t. $y$ for any $\zeta$.*

We then make the following assumptions on the Lipschitzness and bounded variance, as adopted by the existing studies [8, 20, 14] on stochastic bilevel optimization.

**Assumption 2.** *Let $z := (x, y)$. Assume the functions $F(z; \xi)$ and $G(z; \zeta)$ satisfy, for any $\xi$ and $\zeta$,*

a) *$F(z; \xi)$ is $M$-Lipschitz, i.e., for any $z, z'$, $|F(z; \xi) - F(z'; \xi)| \leq M\|z - z'\|$.*

b) *$\nabla F(z; \xi)$ and $\nabla G(z; \zeta)$ are $L$-Lipschitz, i.e., for any $z, z'$,*

$$\|\nabla F(z; \xi) - \nabla F(z'; \xi)\| \leq L\|z - z'\|, \quad \|\nabla G(z; \zeta) - \nabla G(z'; \zeta)\| \leq L\|z - z'\|.$$

c) *$\nabla_x \nabla_y G(z; \zeta)$ is $\tau$-Lipschitz, i.e., for any $z, z'$, $\|\nabla_x \nabla_y G(z; \zeta) - \nabla_x \nabla_y G(z'; \zeta)\| \leq \tau\|z - z'\|$.*

d) *$\nabla_y^2 G(z; \zeta)$ is $\rho$-Lipschitz, i.e., for any $z, z'$, $\|\nabla_y^2 G(z; \zeta) - \nabla_y^2 G(z'; \zeta)\| \leq \rho\|z - z'\|$.*

Note that Assumption 2 also implies that $\mathbb{E}_\xi \|\nabla F(z; \xi) - \nabla f(z)\|^2 \leq M^2$, $\mathbb{E}_\zeta \|\nabla_x \nabla_y G(z; \zeta) - \nabla_x \nabla_y g(z)\|^2 \leq L^2$ and $\mathbb{E}_\zeta \|\nabla_y^2 G(z; \zeta) - \nabla_y^2 g(z)\|^2 \leq L^2$.

**Assumption 3.** *Assume that $\nabla G(z; \xi)$ has bounded variance, i.e., $\mathbb{E}_\xi \|\nabla G(z; \xi) - \nabla g(z)\|^2 \leq \sigma^2$.*

Assumptions 2 and 3 require the Lipschitzness conditions to hold for the gradients and second-order derivatives of the inner and outer objective functions, which further imply the gradient of the outer objective function is bounded. Such assumptions have also been adopted by the existing studies [20, 14, 23, 22, 12] for stochastic bilevel optimization. Furthermore, these assumptions are mild in practice as long as the iterates along practical training paths are bounded. All our experiments

indicate that these iterates are well located in a bounded regime. It is also possible to consider a bilevel problem over a convex compact set, which relaxes the boundedness assumption. By introducing a projection of iterative updates into such a set, our analysis for the unconstrained setting can be extended easily to such a constrained problem.

We next define the $\epsilon$-stationary point for a nonconvex function as the convergence criterion.

**Definition 1.** *We call $\bar{x}$ an $\epsilon$-stationary point for a function $\Phi(x)$ if $\|\nabla\Phi(\bar{x})\|^2 \le \epsilon$.*

## 3.2 Convergence Analysis of MRBO Algorithm

To analyze the convergence of MRBO, bilevel optimization presents two major challenges due to the momentum recursive method in MRBO, beyond the previous studies of momentum in conventional minimization and minimax optimization. (a) Outer-loop updates of bilevel optimization use hypergradients, which involve both the first-order gradient and the Hessian-vector product. Thus, the analysis of the momentum recursive estimator for such a hypergradient is much more complicated than that for the vanilla gradient. (b) Since MRBO applies the momentum-based recursive method to both inner- and outer-loop iterations, the analysis needs to capture the interaction between the inner-loop gradient estimator and the outer-loop hypergradient estimator. Below, we will provide two major properties for MRBO, which develop new analysis for handling the above two challenges.

In the following proposition, we characterize the variance bound for the hypergradient estimator in bilevel optimization, and further use such a bound to characterize the variance of the momentum recursive estimator of the hypergradient.

**Proposition 1.** *Suppose Assumptions 1, 2 and 3 hold and $\eta < \frac{1}{L}$, the hypergradient estimator $\widehat{\nabla}\Phi(x_k; \mathcal{B}_x)$ w.r.t. $x$ based on a minibatch $\mathcal{B}_x$ has bounded variance*

$$\mathbb{E}\|\widehat{\nabla}\Phi(x_k; \mathcal{B}_x) - \overline{\nabla}\Phi(x_k)\|^2 \le G^2, \tag{6}$$

*where $G^2 = \frac{2M^2}{S} + \frac{12M^2L^2\eta^2(Q+1)^2}{S} + \frac{4M^2L^2(Q+2)(Q+1)^2\eta^4\sigma^2}{S}$. Further, let $\bar{\epsilon}_k = v_k - \overline{\nabla}\Phi(x_k)$, where $v_k$ denotes the momentum recursive estimator for the hypergradient. Then the per-iteratioon variance bound of $v_k$ satisfies*

$$\mathbb{E}\|\bar{\epsilon}_k\|^2 \le \mathbb{E}[2\alpha_k^2 G^2 + 2(1-\alpha_k)^2 L_Q^2 \|x_k - x_{k-1}\|^2$$
$$+ 2(1-\alpha_k)^2 L_Q^2 \|y_k - y_{k-1}\|^2 + (1-\alpha_k)^2 \|\bar{\epsilon}_{k-1}\|^2], \tag{7}$$

*where $L_Q^2 = 2L^2 + 4\tau^2\eta^2 M^2(Q+1)^2 + 8L^4\eta^2(Q+1)^2 + 2L^2\eta^4 M^2\rho^2 Q^2(Q+1)^2$.*

The variance bound $G$ of the hypergradient in eq. (6) scales with the number $Q$ of Neumann series terms (i.e., the number of Hessian vectors) and can be reduced by that minibatch size $S$.

Then the bound eq. (7) further captures how the variance $\|\bar{\epsilon}_k\|$ of momentum recursive hypergradient estimator changes after one iteration. Clearly, the term $(1 - \alpha_k)^2 \|\bar{\epsilon}_{k-1}\|^2$ indicates a variance reduction per iteration, and the remain three terms capture the impact of the randomness due to the update in step $k$, including the variance of the stochastic hypergradient estimator $G^2$ (as captured in eq. (6)) and the stochastic update of both variables $x$ and $y$. In particular, the variance reduction term plays a key role in the performance improvement for MRBO over other existing algorithms.

**Proposition 2.** *Suppose Assumptions 1, 2, 3 hold. Let $\eta < \frac{1}{L}$ and $\gamma \le \frac{1}{4L_\Phi\eta_k}$, where $L_\Phi = L + \frac{2L^2+\tau M^2}{\mu} + \frac{\rho LM+L^3+\tau ML}{\mu^2} + \frac{\rho L^2 M}{\mu^3}$. Then, we have*

$$\mathbb{E}[\Phi(x_{k+1})] \le \mathbb{E}[\Phi(x_k)] + 2\eta_k\gamma(L'^2\|y_k - y^*(x_k)\|^2 + \|\bar{\epsilon}_k\|^2 + C_Q^2) - \frac{1}{2\gamma\eta_k}\|x_{k+1} - x_k\|^2,$$

*where $C_Q = \frac{(1-\eta\mu)^{Q+1}ML}{\mu}$, $L'^2 = \max\{(L + \frac{L^2}{\mu} + \frac{M\tau}{\mu} + \frac{LM\rho}{\mu^2})^2, L_Q^2\}$.*

Proposition 2 characterizes how the objective function value decreases (i.e., captured by $\mathbb{E}[\Phi(x_{k+1})] - \mathbb{E}[\Phi(x_k)]$) due to one-iteration update $\|x_{k+1} - x_k\|^2$ of variable $x$ (last term in the bound). Such a value reduction is also affected by the tracking error $\|y_k - y^*(x_k)\|^2$ of the variable $y$ (i.e., $y_k$ does not equal the desirable $y^*(x_k)$), the variance $\|\bar{\epsilon}_k\|^2$ of momentum recursive hypergradient estimator, and the Hessian inverse approximation error $C_Q$ w.r.t. hypergradient.

Based on Propositions 1 and 2, we next characterize the convergence of MRBO.

**Theorem 1.** *Apply MRBO to solve the problem eq. (1). Suppose Assumptions 1, 2, and 3 hold. Let hyperparameters $c_1 \geq \frac{2}{3d^3} + \frac{9\lambda\mu}{4}, c_2 \geq \frac{2}{3d^3} + \frac{75L'^2\lambda}{2\mu}, m \geq \max\{2, d^3, (c_1 d)^3, (c_2 d)^3\}, y_1 = y^*(x_1), \eta < \frac{1}{L}, 0 \leq \lambda \leq \frac{1}{6L}, 0 \leq \gamma \leq \min\{\frac{1}{4L_\Phi \eta_K}, \frac{\lambda\mu}{\sqrt{150L'^2 L^2/\mu^2 + 8\lambda\mu(L_Q^2 + L^2)}}\}$. Then, we have*

$$\frac{1}{K} \sum_{k=1}^{K} \left( \frac{L'^2}{4} \|y^*(x_k) - y_k\|^2 + \frac{1}{4}\|\bar{\epsilon}_k\|^2 + \frac{1}{4\gamma^2 \eta_k^2}\|x_{k+1} - x_k\|^2 \right) \leq \frac{M'}{K}(m + K)^{1/3}, \quad (8)$$

*where $L'^2$ is defined in Proposition 2, and $M' = \frac{\Phi(x_1) - \Phi^*}{\gamma d} + \left( \frac{2G^2(c_1^1 + c_2^2)d^2}{\lambda\mu} + \frac{2C_Q^2 d^2}{\eta_K^2} \right) \log(m + K) + \frac{2G^2}{S\lambda\mu d\eta_0}$.*

Theorem 1 captures the simultaneous convergence of the variables $x_k, y_k$ and $\|\bar{\epsilon}_k\|$: the tracking error $\|y^*(x_k) - y_k\|$ converges to zero, and the variance $\|\bar{\epsilon}_k\|$ of the momentum recursive hypergradient estimator reduces to zero, both of which further facilitate the convergence of $x_k$ and the algorithm.

By properly choosing the hyperparameters in Algorithm 1 to satisfy the conditions in Theorem 1, we obtain the following computational complexity for MRBO.

**Corollary 1.** *Under the same conditions of Theorem 1 and choosing $K = \mathcal{O}(\epsilon^{-1.5}), Q = \mathcal{O}(\log(\frac{1}{\epsilon}))$, MRBO in Algorithm 1 finds an $\epsilon$-stationary point with the gradient complexity of $\mathcal{O}(\epsilon^{-1.5})$ and the (Jacobian-) Hessian-vector complexity of $\widetilde{\mathcal{O}}(\epsilon^{-1.5})$.*

As shown in Corollary 1, MRBO achieves the computational complexity of $\widetilde{\mathcal{O}}(\epsilon^{-1.5})$, which outperforms all existing stochastic bilevel algorithms by a factor of $\widetilde{\mathcal{O}}(\epsilon^{-0.5})$ (see Table 1). Further, this also achieves the best known complexity of $\widetilde{\mathcal{O}}(\epsilon^{-1.5})$ for vanilla nonconvex optimization via first-order stochastic algorithms. As far as we know, this is the first result to demonstrate the improved performance of single-loop recursive momentum over SGD-type updates for bilevel optimization.

### 3.3 Convergence Analysis of VRBO Algorithm

To analyze the convergence of VRBO, we need to first characterize the statistical properties of the hypergradient estimator, in which all the gradient, Jacobian-vector, and Hessian-vector have recursive variance reduction forms. We then need to characterize how the inner-loop tracking error affects the outer-loop hypergradient estimation error in order to establish the overall convergence. The complication in the analysis is mainly due to the hypergradient in bilevel optimization, which does not exist in the previous studies of variance reduction in conventional minimization and minimax optimization. Below, we provide two properties of VRBO for handling the aforementioned challenges.

In the following proposition, we characterize the variance of the hypergradient estimator, and further use such a bound to characterize the cumulative variances of both the hypergradient and inner-loop gradient estimators based on the recursive variance reduction technique over all iterations.

**Proposition 3.** *Suppose Assumptions 1, 2, 3 hold. Let $\eta < \frac{1}{L}$. Then the hypergradient estimator $\widehat{\nabla}\Phi(x_k, y_k; \mathcal{S}_1)$ defined in eq. (5) w.r.t. $x$ has bounded variance as*

$$\mathbb{E}\|\widehat{\nabla}\Phi(x_k, y_k; \mathcal{S}_1) - \overline{\nabla}\Phi(x_k)\|^2 \leq \frac{\sigma'^2}{S_1}, \quad (9)$$

*where $\sigma'^2 = 2M^2 + 28L^2 M^2 \eta^2 (Q+1)^2$. Let $\Delta_k = \mathbb{E}(\|v_k - \overline{\nabla}\Phi(x_k)\|^2 + \|u_k - \nabla_y g(x_k, y_k)\|^2)$, where $v_k$ and $u_k$ denote the recursive variance reduction estimators for hypergradient and inner-loop gradient respectively. Then, the cumulative variance of $v_k$ and $u_k$ is bounded by*

$$\sum_{k=0}^{K-1} \Delta_k \leq \frac{4\sigma'^2 K}{S_1} + 22\alpha^2 L_Q^2 \sum_{k=0}^{K-2} \mathbb{E}\|v_k\|^2 + \frac{4}{3}\mathbb{E}\|\nabla_y g(x_0, y_0)\|^2. \quad (10)$$

As shown in eq. (9), the variance bound of the hypergradient estimator increases with the number $Q$ of Hessian-vector products for approximating the Hessian inverse and can be reduced by the batch size $S_1$. Then eq. (10) further provides an upper bound on the cumulative variance $\sum_{k=0}^{K-1} \Delta_k$ of the recursive hypergradient estimator and inner-loop gradient estimator.

**Proposition 4.** *Suppose Assumptions 1, 2, 3 hold. Let $\eta < \frac{1}{L}$. Then, we have*

$$\mathbb{E}[\Phi(x_{k+1})] \leq \mathbb{E}[\Phi(x_k)] + \frac{\alpha L'^2}{\mu^2}\mathbb{E}\|\nabla_y g(x_k, y_k)\|^2 + \alpha\mathbb{E}\|\widetilde{\nabla}\Phi(x_k) - v_k\|^2 - (\frac{\alpha}{2} - \frac{\alpha^2}{2}L_\Phi)\mathbb{E}\|v_k\|^2,$$

*where $L'^2 = (L + \frac{L^2}{\mu} + \frac{M\tau}{\mu} + \frac{LM\rho}{\mu^2})^2$ and $\widetilde{\nabla}\Phi(x_k)$ takes a form of*

$$\widetilde{\nabla}\Phi(x_k) = \nabla_x f(x_k, y_k) - \nabla_x\nabla_y g(x_k, y_k)[\nabla_y^2 g(x_k, y_k)]^{-1}\nabla_y f(x_k, y_k). \tag{11}$$

Proposition 4 characterizes how the objective function value decreases (i.e., captured by $\mathbb{E}[\Phi(x_{k+1})] - \mathbb{E}[\Phi(x_k)]$) due to one iteration update $\|v_k\|^2$ of variable $x$ (last term in the bound). Such a value reduction is also affected by the moments of gradient w.r.t. $y$ and the variance of recursive hypergradient estimator.

Based on Propositions 3 and 4, we next characterize the convergence of VRBO.

**Theorem 2.** *Apply VRBO to solve the problem eq. (1). Suppose Assumptions 1, 2, 3 hold. Let $\alpha = \frac{1}{20L_m^3}, \beta = \frac{2}{13L_Q}, \eta < \frac{1}{L}, S_2 \geq 2(\frac{L}{\mu}+1)L\beta, m = \frac{16}{\mu\beta} - 1, q = \frac{\mu L\beta S_2}{\mu+L}$ where $L_m = \max\{L_Q, L_\Phi\}$. Then, we have*

$$\frac{1}{K}\sum_{k=0}^{K-1}\mathbb{E}\|\nabla\Phi(x_k)\|^2 \leq \mathcal{O}(\frac{Q^4}{K} + \frac{Q^6}{S_1} + Q^4(1-\eta\mu)^{2Q}). \tag{12}$$

Theorem 2 shows that VRBO converges sublinearly w.r.t. the number $K$ of iterations with the convergence error consisting of two terms. The first error term $\frac{Q^6}{S_1}$ is caused by the minibatch gradient and hypergradient estimation at outer loops and can be reduced by increasing the batch size $S_1$ (in fact, $Q$ scales only logarithmically with $S_1$). The second error term $Q^4(1-\eta\mu)^{2Q}$ is due to the approximation error of the Hessian-vector type of hypergradient estimation, which decreases exponentially fast w.r.t. $Q$. By properly choosing the hyperparameters in Algorithm 2, we obtain the following complexity result for VRBO.

**Corollary 2.** *Under the same conditions of Theorem 2, choose $S_1 = \mathcal{O}(\epsilon^{-1}), S_2 = \mathcal{O}(\epsilon^{-0.5}), Q = \mathcal{O}(\log(\frac{1}{\epsilon^{0.5}})), K = \mathcal{O}(\epsilon^{-1})$. Then, VRBO finds an $\epsilon$-stationary point with the gradient complexity of $\widetilde{\mathcal{O}}(\epsilon^{-1.5})$ and Hessian-vector complexity of $\widetilde{\mathcal{O}}(\epsilon^{-1.5})$.*

Similarly to MRBO, Corollary 2 indicates that VRBO also outperforms all existing stochastic algorithms for bilevel optimization by a factor of $\widetilde{\mathcal{O}}(\epsilon^{-0.5})$ (see Table 1). Further, although MRBO and VRBO achieve the same theoretical computational complexity, VRBO empirically performs much better than MRBO (as well as other single-loop momentum-based algorithms MSTSA [22], STABLE [2], and SEMA [11]), as will be shown in Section 4.

We note that although our theory requires $Q$ to scale as $\mathcal{O}(\log(\frac{1}{\epsilon^{0.5}}))$, a very small $Q$ is sufficient to attain a fast convergence speed in experiments. For example, we choose $Q = 3$ in our hyper-cleaning experiments as other benchmark algorithms such as AID-FP, reverse, and stocBiO.

## 4 Experiments

In this section, we compare the performances of our proposed VRBO and MRBO algorithms with the following bilevel optimization algorithms: AID-FP [10], reverse [6] (both are double-loop deterministic algorithms), BSA [8] (double-loop stochastic algorithm), MSTSA [22] and SUSTAIN [23] (single-loop stochastic algorithms), STABLE [2] (single-loop stochastic algorithm with Hessian inverse computations), and stocBiO [20] (double-loop stochastic algorithm). SEMA [11] is not included in the list because it performs similarly to SUSTAIN. RSVRB [12] is not included since it performs similarly to STABLE. Our experiments are run over a hyper-cleaning application on MNIST[3]. We provide the detailed experiment specifications in Appendix B.

As shown in Figure 1 (a) and (b), the convergence rate (w.r.t. running time) of our VRBO and the SGD-type stocBiO converge much faster than other algorithms in comparison. Between VRBO and stocBiO, they have comparable performance, but our VRBO achieves a lower training loss as well as

---

[3]The experiments on CIFAR10 are still ongoing.

a more stable convergence. Further, our VRBO converges significantly faster than all single-loop momentum-based methods. This provides some evidence on the advantage of double-loop algorithms over single-loop algorithms for bilevel optimization. Moreover, our MRBO achieves the fastest convergence rate among all single-loop momentum-based algorithms, which is in consistent with our theoretical results. In Figure 1 (c), we compare our algorithms MRBO and VRBO with three momentum-based algorithms, i.e., MSTSA, STABLE, and SUSTAIN, where SUSTAIN (proposed in the concurrent work [23]) achieves the same theoretical complexity as our MRBO and VRBO. However, it can be seen that MRBO and VRBO are significantly faster than the other three algorithms.

All three plots suggest an interesting observation that **double-loop** algorithms tend to converge faster than **single-loop** algorithms as demonstrated by (i) double-loop VRBO performs the best among all algorithms; and (ii) double-loop SGD-type stocBiO, GD-type reverse and AID-FP perform even better than single-loop momentum-accelerated stochastic algorithm MRBO; and (iii) double-loop SGD-type BSA (with single-sample updates) converges faster than single-loop momentum-accelerated stochastic MSTSA, STABLE and SUSTAIN (with single-sample updates). Such a phenomenon has been observed only in bilevel optimization (to our best knowledge), and occurs oppositely in minimization and minimax problems, where single-loop algorithms substantially outperform double-loop algorithms. The reason for this can be that the hypergradient estimation at the outer-loop in bilevel optimization is very sensitive to the inner-loop output. Thus, for each outer-loop iteration, sufficient inner-loop iterations in the double-loop structure provides a much more accurate output close to $y^*(x)$ than a single inner-loop iteration, and thus helps to estimate a more accurate hypergradient in the outer loop. This further facilitates better outer-loop iterations and yields faster overall convergence.

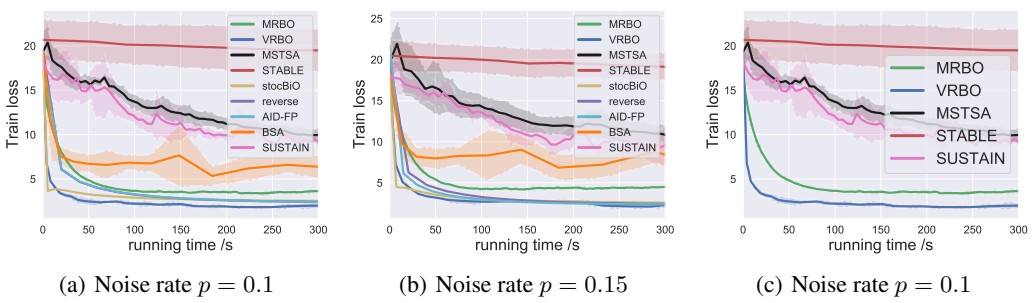

(a) Noise rate $p = 0.1$         (b) Noise rate $p = 0.15$         (c) Noise rate $p = 0.1$

Figure 1: training loss v.s. running time.

## 5    Conclusion

In this paper, we proposed two novel algorithms MRBO and VRBO for the nonconvex-strongly-convex bilevel stochastic optimization problem, and showed that their computational complexities outperform all existing algorithms order-wise. In particular, MRBO is the first momentum algorithm that exhibits the order-wise improvement over SGD-type algorithms for bilevel optimization, and VRBO is the first that adopts the recursive variance reduction technique to accelerate bilevel optimization. Our experiments demonstrate the superior performance of these algorithms, and further suggest that the double-loop design may be more suitable for bilevel optimization than the single-loop structure. We anticipate that our analysis can be applied to studying bilevel problems under various other loss geometries. We also hope that our study can motivate further comparison between double-loop and single-loop algorithms in bilevel optimization.

## Acknowledgements

The work was supported in part by the U.S. National Science Foundation under the grants ECCS-2113860, DMS-2134145 and CNS-2112471.

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
