# Supplementary Materials

## A   Hessian Vector Implementation

In this section, we provide an algorithm (see Algorithm 3) for computing the hypergradient estimator in eq. (4) in MRBO by **Hessian vectors** rather than **Hessians**, in order to reduce the memory and computational cost.

---

**Algorithm 3** Hessian Vector Implementation for Computing Hypergradient Estimator in eq. (4)

---

1: **Input:** Hessian Estimation Number $Q$, Samples $\mathcal{B}_x$, Hyperparameter $\eta$,
2: Compute $\nabla_x F(x, y; \mathcal{B}_F)$, $r_0 = \nabla_y F(x, y; \mathcal{B}_F)$, $\nabla_y G(x, y; \mathcal{B}_G)$
3: **for** $q = 0, 1, \ldots, Q - 1$ **do**
4:     $G_{q+1} = (y - \eta \nabla_y G(x, y; \mathcal{B}_{Q-q})) r_q$
5:     $r_{q+1} = \partial(G_{q+1})/\partial y$      note: $\partial(G_{q+1})/\partial y = r_q - \eta \nabla_y^2 G(x, y; \mathcal{B}_{Q-q}) r_q$
6: **end for**
7: $M_Q = \eta \sum_{q=0}^{Q} r_q$
8: Return $\nabla_x F(x, y; \mathcal{B}_F) - \partial(\nabla_y G(x, y; \mathcal{B}_G) M_Q)/\partial x$

---

As shown in line 5 of Algorithm 3, instead of updating $r_{q+1} = r_q - \eta \nabla_y^2 G(x, y; \mathcal{B}_{Q-q}) r_q$ by directly computing Hessian $\nabla_y^2 G(x, y; \mathcal{B}_{Q-q})$, we choose to compute the Hessian-vector product via $r_{q+1} = \partial(G_{q+1})/\partial y$. A similar implementation is applied to compute the Jacobian vector $\partial(\nabla_y G(x, y; \mathcal{B}_G) M_Q)/\partial x$ in line 8. Note that both lines 5 and 8 can apply automatic differentiation function *torch.grad()* for easy implementation. In this way, we compute the hypergradient estimator in eq. (4) recursively (see lines 3-6 in Algorithm 3) via Hessian-vector products without computing Hessian explicitly.

## B   Specifications of Experiments

We compare our proposed algorithms MRBO and VRBO with other benchmarks including stocBiO [20], reverse [6], AID-FP [10], BSA [8], MSTSA [22], STABLE [2] and SUSTAIN [23] on the hyper-cleaning problem [34] with MNIST dataset [25]. The formulation of data hyper-cleaning is given below:

$$\min_{\lambda} \mathbb{E}[\mathcal{L}_\mathcal{V}(\lambda, w^*)] = \frac{1}{|S_\mathcal{V}|} \sum_{(x_i, y_i) \in S_\mathcal{V}} L_{CE}((w^*)^T x_i, y_i)$$

$$\text{s.t.} \quad w^* = \arg\min_w \mathcal{L}(\lambda, w) := \frac{1}{|S_\mathcal{T}|} \sum_{(x_i, y_i) \in S_\mathcal{T}} \sigma(\lambda_i) L_{CE}(w^T x_i, y_i) + C\|w\|^2,$$

where $L_{CE}$ denotes the cross-entropy loss, $S_\mathcal{T}$ and $S_\mathcal{V}$ denote the training data and the validation data, respectively, $\lambda = \{\lambda_i\}_{i \in S_\mathcal{T}}$ and $C$ are the regularization parameters, and $\sigma(\cdot)$ denotes the sigmoid function. In experiment, we set $C = 0.001$ and fix the size of the training data $S_\mathcal{V}$ and validation data $S_\mathcal{T}$ as 20000 and 5000, respectively. Furthermore, we use 10000 images for testing, which follows the setting in [20]. We use the Hessian-vector based algorithm (Algorithm 3) for computing the hypergradient estimator, where we set $Q = 3$ and $\eta = 0.5$. For stochastic algorithms including MRBO, VRBO, stocBiO, we set the batchsize to be 1000 for both training and validation procedures. For VRBO, we set the inner batchsize to be 500 and the period $q$ to be 3. For the double-loop algorithms, we fine tune the number of inner-loop steps and set it to be 200 for the stocBiO, AID-FP, BSA and reverse algorithms for the best performance, and set it to be 20 for VRBO for the best performance. To set the outer-loop and inner-loop stepsizes, we use the training loss as the metric and apply the standard grid search with the stepsizes $\lambda$, $\gamma$, $\alpha$ and $\beta$ all chosen from the interval [1e-3,1]. We then select those that yield the best convergence performance. Thus, we set 0.1 as the stepsize for all algorithms except SUSTAIN and STABLE. For SUSTAIN, the inner-loop stepsize is set to be 0.03 and outer-loop stepsize is set to be 0.1, and for STABLE, inner-loop and outer-loop stepsizes are set to be 0.01 and 1e-10, respectively, because these algorithms are not stable with larger stepsizes. Our experimental implementations are based on the implementation of stocBiO in [20], which is under MIT License. Futhermore, all results are repeated with 5 random seeds and

we use iMac with 3.8GHz Quad-Core Intel Core i5 CPU and 32 GB 2400 MHz DDR4 for training without the requirement of GPU. However, our code supports GPU cluster training.

## B.1  Additional Experiments of Hyper-cleaning

In this subsection, we include extra experiments to further validate our theoretical results and understand the VRBO algorithm.

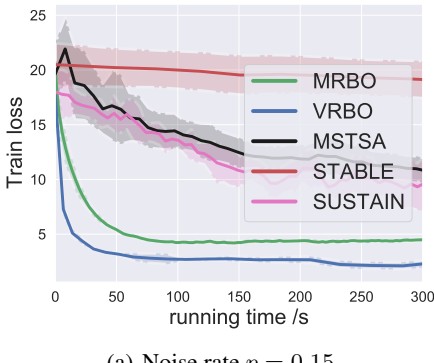

(a) Noise rate $p = 0.15$

Figure 2: training loss v.s. running time.

In Figure 2, we compare our algorithms MRBO and VRBO with three momentum-based algorithms, i.e., MSTAS, STABLE, and SUSTAIN, under the noise rate $p = 0.15$, which is a scenario in addition to the experiment provided in Figure 1 (c) of the main part under the noise rate $p = 0.1$. It is clear that our algorithms MRBO and VRBO achieve the lowest training loss and converge fastest among all momentum-based algorithms.

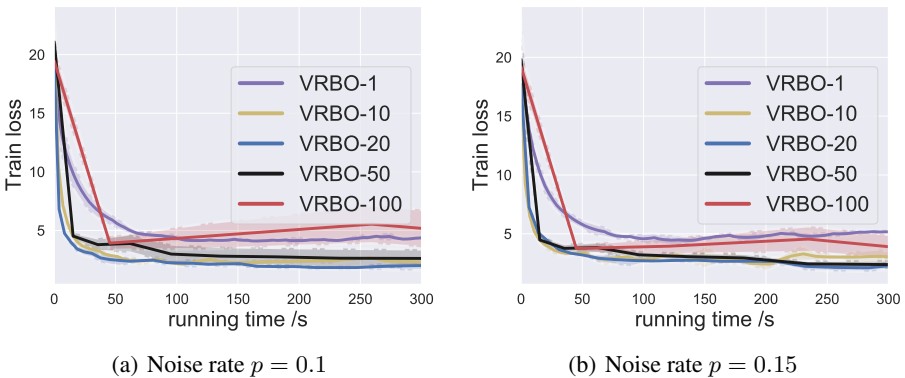

(a) Noise rate $p = 0.1$          (b) Noise rate $p = 0.15$

Figure 3: training loss v.s. running time.

The next experiment focuses on the double-loop algorithm VRBO and studies how the number $m$ of inner-loop steps affects its performance. In Figure 3 (a) and (b), we compare VRBO among five choices of $m \in \{1, 10, 20, 50, 100\}$, where VRBO-$m$ in the legend indicates that the inner-loop of VRBO takes $m$ steps. It can be observed that as $m$ increases from 1, VRBO becomes more stable and achieves lower training loss until $m = 20$. Beyond this point, as $m$ further increases, the performance of VRBO becomes worse with higher final training loss and lower stability. This can be explained by two reasons: (i) the accuracy of the inner-loop output and (ii) the accuracy of the variance-reduced gradient estimator. By the formulation of bilevel optimization, at each outer-loop step $k$, it is desirable that the inner loop obtains $y_k$ as close as possible to the optimal point $y^*(x_k) = \arg\min_y g(x_k, y)$. Hence, taking more inner-loop steps (i.e., as $m$ increases) helps to obtain more accurate $y_k$. Further, increasing $m$ allows the large-batch gradient estimator to benefit more steps of gradient estimators in the inner loop via variance reduction, and hence improves the computational efficiency. Both reasons explain that the overall performance of VRBO gets better as $m$ increases from $m = 1$ to $m = 20$. On

the other hand, when $m$ is large enough (i.e., $m = 20$ in our plots), the inner-loop can already provide a sufficiently accurate $y_k$. Then further increasing $m$ will cause unnecessary inner-loop iterations and hurt the computational efficiency. Moreover, larger $m$ causes the variance-reduced gradient estimators in the later stage of the inner loop becomes less accurate. Thus, the overall convergence of VRBO becomes slower and less stable.

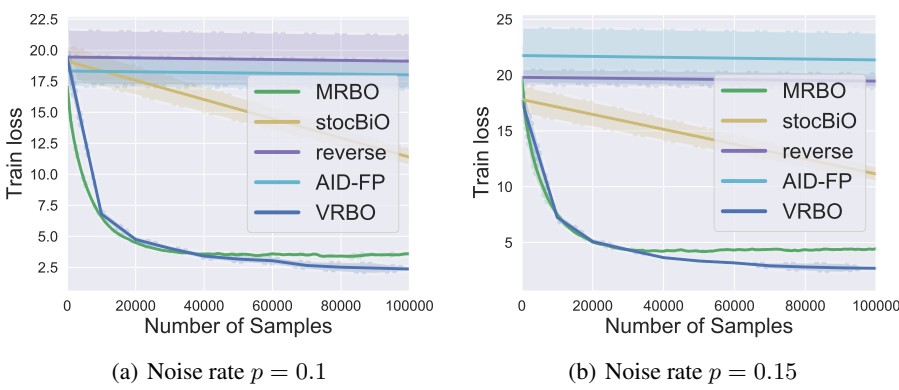

(a) Noise rate $p = 0.1$        (b) Noise rate $p = 0.15$

Figure 4: training loss v.s. number of samples.

In Figure 4, we further compare our algorithms with other **batch-sample** based algorithms in terms of the training performance versus the number of samples required. It can be seen that MRBO and VRBO are much more sample efficient in training compared with stocBiO and the GD-based algorithms reverse and AID-FP.

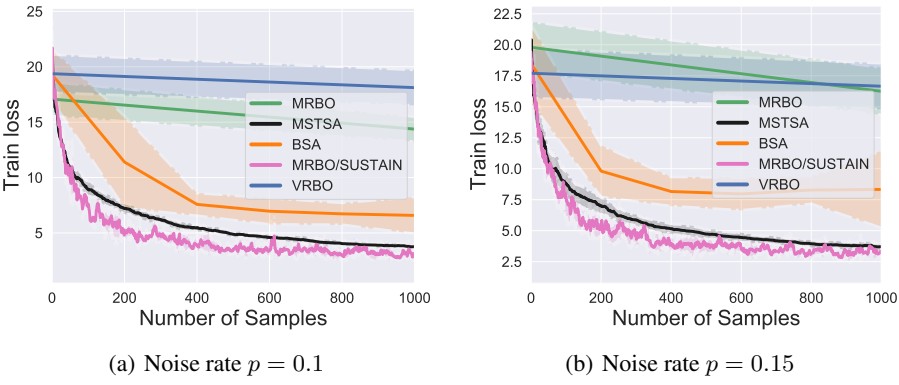

(a) Noise rate $p = 0.1$        (b) Noise rate $p = 0.15$

Figure 5: training loss v.s. number of samples.

We also compare our algorithms with other **single-sample** based algorithms w.r.t. the number of samples in Figure 5. It can be seen that single-sample based algorithms are more sample efficient than MRBO and VRBO. This is because single-sample based algorithms update each parameter using a single sample, whereas batch-sample based algorithms update each parameter using a batch of samples. As a result, single-sample based algorithms enable a larger parameter update per sample, and hence achieve a higher sample efficiency. It is worthy to mention that our MRBO can be implemented in a single-sample fashion, which then becomes the same algorithm as the concurrently proposed algorithm SUSTAIN. However, compared to the sample efficiency, we believe that the execution time (under the same computing resource) is a more reasonable measure of the computational efficiency of bilevel algorithms. This is because the minibatch computation are more preferred and efficient than the single-sample computation in existing deep learning platforms such as PyTorch. Thus, as demonstrated in our Figure 1, batch-sample based algorithms converge much faster than single-sample based algorithms w.r.t. running time.

## B.2 Experiments of Logistic Regression

We further conduct the experiment on the logistic regression problem over the 20 Newsgroup dataset [10]. The objective function is given by:

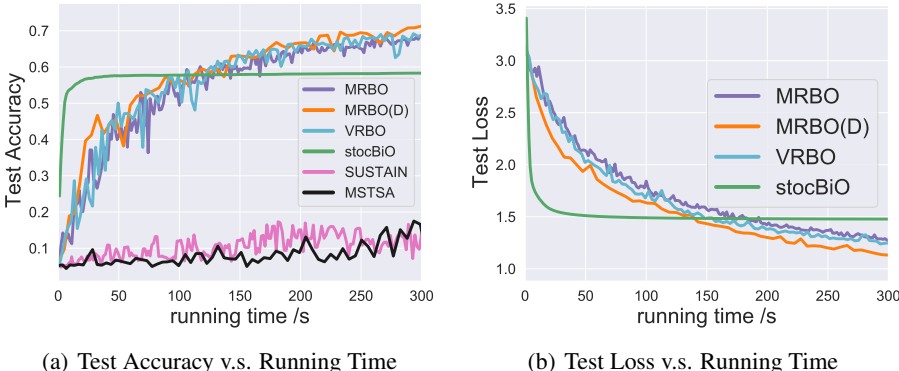

(a) Test Accuracy v.s. Running Time      (b) Test Loss v.s. Running Time

Figure 6: test accuracy or test loss v.s. running time (Batchsize=100)

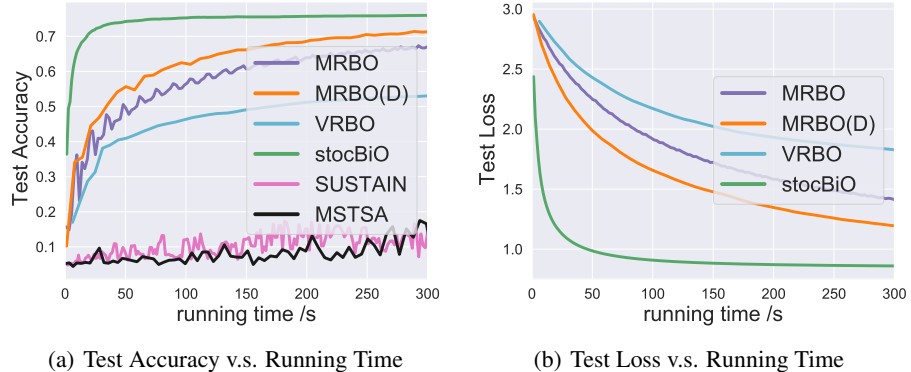

(a) Test Accuracy v.s. Running Time      (b) Test Loss v.s. Running Time

Figure 7: test accuracy or test loss v.s. running time (Batchsize=1000)

$$\min_{\lambda} \mathbb{E}[\mathcal{L}_{\mathcal{V}}(\lambda, w^*)] = \frac{1}{|S_{\mathcal{V}}|} \sum_{(x_i, y_i) \in S_{\mathcal{V}}} L_{CE}((w^*)^T x_i, y_i)$$

$$\text{s.t.} \quad w^* = \arg\min_{w} \mathcal{L}(\lambda, w) := \frac{1}{|S_{\mathcal{T}}|} \sum_{(x_i, y_i) \in S_{\mathcal{T}}} L_{CE}(w^T x_i, y_i) + \frac{1}{cp} \sum_{i=1}^{c} \sum_{j=1}^{p} \exp(\lambda_j) w_{ij}^2,$$

where $L_{CE}$ denotes the cross-entropy loss, $S_{\mathcal{T}}$ and $S_{\mathcal{V}}$ denote the training and validation datasets, respectively. In the experiment, we follow the setting for stocBiO in [20] and set $\eta = 0.5$ and $Q = 10$ for the hypergradient estimation. Besides, we apply the standard grid search for the inner- and outer-loop stepsizes for all algorithms. Thus, we set inner- and outer-loop stepsizes as 100 for stocBiO, inner- and outer-loop stepsizes as 30 for MRBO, VRBO, SUSTAIN and MSTSA. Following the setting in stocBiO, we set inner-loop steps as 10 for stocBiO. For VRBO, we set the period $q$ as 2 and inner-loop steps as 3 for the best performance. We also conduct MRBO in a double loop fashion and call it as MRBO(D), where we apply the inner update procedure 10 times per epoch.

In Figure 6, we set the batchsize of all stochastic algorithms to 100. It can be seen that although stocBiO achieves the fastest initial convergence rate, both MRBO and VRBO reach a higher accuracy than stocBiO due to more accurate hypergradient estimation. It can be also seen that our double-loop MRBO(D) achieves the highest accuracy, whereas single-loop SUSTAIN and MSTSA algorithms do not converge well. This demonstrates the advantage of double-loop updates over single-loop updates.

In Figure 7, we choose a larger batchsize of 1000 for all algorithms. We note that double-loop algorithms stocBiO and MRBO(D) still outperform other single-loop algorithms significantly, and stocBiO achieves the best test accuracy due to a more accurate gradient estimation.

# C    Proof of Theorem 1

## C.1    Proof of Supporting Lemmas (Propositions 1 and 2 Correspond to Lemmas 4 and 8)

For notation simplification, we define the following:

$$V_{Qk} = \eta \sum_{q=-1}^{Q-1} \prod_{j=Q-q}^{Q} (I - \eta \nabla_y^2 G(x_k, y_k; \mathcal{B}_j)) \nabla_y F(x_k, y_k; \mathcal{B}_F). \tag{13}$$

Firstly, we characterize the variance of $V_{Qk}$ in the following lemma.

**Lemma 1.** *Suppose Assumptions 2, 3 hold. Let* $\eta < \frac{1}{L}$. *Then, we have*

$$\mathbb{E}\|V_{Qk} - \mathbb{E}[V_{Qk}]\|^2 \leq \frac{2\eta^2 M^2 (Q+1)^2}{S} + \frac{M^2 (Q+2)(Q+1)^2 \eta^2 \sigma^2}{2S},$$

*where* $V_{Qk}$ *is defined in eq.* (13).

*Proof.* Based on the form of $V_{Qk}$, we have

$$\mathbb{E}\|V_{Qk} - \mathbb{E}[V_{Qk}]\|^2$$

$$\overset{(i)}{=} \eta^2 \mathbb{E} \left\| \sum_{q=0}^{Q} (I - \eta \nabla_y^2 g(x_k, y_k))^q \nabla_y f(x_k, y_k) - \sum_{q=0}^{Q} (I - \eta \nabla_y^2 g(x_k, y_k))^q \nabla_y F(x_k, y_k; \mathcal{B}_F) \right.$$

$$+ \sum_{q=0}^{Q} (I - \eta \nabla_y^2 g(x_k, y_k))^q \nabla_y F(x_k, y_k; \mathcal{B}_F)$$

$$\left. - \sum_{q=-1}^{Q-1} \prod_{j=Q-q}^{Q} (I - \eta \nabla_y^2 G(x_k, y_k; \mathcal{B}_j)) \nabla_y F(x_k, y_k; \mathcal{B}_F) \right\|^2$$

$$\overset{(ii)}{\leq} 2\eta^2 M^2 \mathbb{E} \left\| \sum_{q=0}^{Q} (I - \eta \nabla_y^2 g(x_k, y_k))^q - \sum_{q=-1}^{Q-1} \prod_{j=Q-q}^{Q} (I - \eta \nabla_y^2 G(x_k, y_k; \mathcal{B}_j)) \right\|^2$$

$$+ \frac{2\eta^2 M^2 (Q+1)^2}{S}$$

$$\leq 2\eta^2 M^2 (Q+1) \mathbb{E} \sum_{q=0}^{Q} \left\| (I - \eta \nabla_y^2 g(x_k, y_k))^q - \prod_{j=Q+1-q}^{Q} (I - \eta \nabla_y^2 G(x_k, y_k; \mathcal{B}_j)) \right\|^2$$

$$+ \frac{2\eta^2 M^2 (Q+1)^2}{S}$$

$$\leq \frac{2\eta^2 M^2 (Q+1)^2}{S} + 2\eta^2 M^2 (Q+1) \mathbb{E} \sum_{q=0}^{Q} \left\| (I - \eta \nabla_y^2 g(x_k, y_k))^q \right.$$

$$- (I - \eta \nabla_y^2 g(x_k, y_k))^{q-1}(I - \eta \nabla_y^2 G(x_k, y_k; \mathcal{B}_Q))$$

$$\left. + (I - \eta \nabla_y^2 g(x_k, y_k))^{q-1}(I - \eta \nabla_y^2 G(x_k, y_k; \mathcal{B}_Q)) - \prod_{j=Q+1-q}^{Q} (I - \eta \nabla_y^2 G(x_k, y_k; \mathcal{B}_j)) \right\|^2$$

$$\overset{(iii)}{\leq} 2\eta^2 M^2 (Q+1) \mathbb{E} \sum_{q=0}^{Q} (q+1) \left\| (I - \eta \nabla_y^2 G(x_k, y_k; \mathcal{B}_q)) - (I - \eta \nabla_y^2 g(x_k, y_k)) \right\|^2$$

$$+ \frac{2\eta^2 M^2 (Q+1)^2}{S}$$

$$\overset{(iv)}{\leq} \frac{2\eta^2 M^2 (Q+1)^2}{S} + \frac{M^2(Q+2)(Q+1)^2\eta^4\sigma^2}{S},$$

where $(i)$ follows from the fact that $\mathbb{E}[V_{Qk}] = \eta\sum_{q=0}^{Q}(I-\eta\nabla_y^2 g(x_k,y_k))^q\nabla_y f(x_k,y_k)$, $(ii)$ follows from Assumption 2 and the fact that $\|I-\eta\nabla_y^2 g(x_k,y_k)\| \leq 1$, $(iii)$ follows from the facts that $\|I-\eta\nabla_y^2 G(x_k,y_k;\mathcal{B}_j)\| \leq 1$ and $\|I-\eta\nabla_y^2 g(x_k,y_k)\| \leq 1$, and $(iv)$ follows from Assumptions 2 and 3. Then, the proof is complete. $\qquad\square$

Futhermore, we characterize the Lipschitz property of $V_{Qk}$ in the following lemma.

**Lemma 2.** *Suppose Assumption 2 holds. Let $\eta < \frac{1}{L}$. Then, we have*

$$\|V_{Qk} - V_{Q(k-1)}\|^2 \leq \left(\frac{M^2 Q^2(Q+1)^2\eta^4\rho^2}{2} + 2\eta^2 L^2(Q+1)^2\right)\|z_k - z_{k-1}\|^2, \quad (14)$$

*where $V_{Qk}$ is defined in eq. (13).*

*Proof.* Based on the form of $V_{Qk}$, we have

$$\|V_{Qk} - V_{Q(k-1)}\|^2$$

$$\overset{(i)}{\leq} \eta^2 \left\| \sum_{q=-1}^{Q-1}\prod_{j=Q-q}^{Q}(I-\eta\nabla_y^2 G(x_k,y_k;\mathcal{B}_j))\nabla_y F(x_k,y_k;\mathcal{B}_F)\right.$$
$$\left. - \sum_{q=-1}^{Q-1}\prod_{j=Q-q}^{Q}(I-\eta\nabla_y^2 G(x_{k-1},y_{k-1};\mathcal{B}_j))\nabla_y F(x_{k-1},y_{k-1};\mathcal{B}_F)\right\|^2$$

$$\leq \eta^2\left(2\|\nabla_y F(x_k,y_k;\mathcal{B}_F)\|^2\left\|\sum_{q=-1}^{Q-1}\prod_{j=Q-q}^{Q}(I-\eta\nabla_y^2 G(x_k,y_k;\mathcal{B}_j))\right.\right.$$
$$\left.\left.- \sum_{q=-1}^{Q-1}\prod_{j=Q-q}^{Q}(I-\eta\nabla_y^2 G(x_{k-1},y_{k-1};\mathcal{B}_j))\right\|^2\right)$$

$$+ 2\|\nabla_y F(x_k,y_k;\mathcal{B}_F) - \nabla_y F(x_{k-1},y_{k-1};\mathcal{B}_F)\|^2\left\|\sum_{q=-1}^{Q-1}\prod_{j=Q-q}^{Q}(I-\eta\nabla_y^2 G(x_{k-1},y_{k-1};\mathcal{B}_j))\right\|^2$$

$$\overset{(ii)}{\leq}\eta^2\left(2M^2\left\|\sum_{q=-1}^{Q-1}\prod_{j=Q-q}^{Q}(I-\eta\nabla_y^2 G(x_k,y_k;\mathcal{B}_j)) - \sum_{q=-1}^{Q-1}\prod_{j=Q-q}^{Q}(I-\eta\nabla_y^2 G(x_{k-1},y_{k-1};\mathcal{B}_j))\right\|^2\right.$$

$$\left. + 2L^2(Q+1)^2\|z_k - z_{k-1}\|^2\right)$$

$$\overset{(iii)}{\leq}2\eta^2 M^2\left(\sum_{q=0}^{Q}\left\|\prod_{j=Q+1-q}^{Q}(I-\eta\nabla_y^2 G(x_k,y_k;\mathcal{B}_j)) - \prod_{j=Q+1-q}^{Q}(I-\eta\nabla_y^2 G(x_{k-1},y_{k-1};\mathcal{B}_j))\right\|\right)^2$$

$$+ 2\eta^2 L^2(Q+1)^2\|z_k - z_{k-1}\|^2$$

$$\overset{(iv)}{\leq}2\eta^2 M^2\left(\sum_{q=0}^{Q}q\eta\rho\|z_k - z_{k-1}\|\right)^2 + 2\eta^2 L^2(Q+1)^2\|z_k - z_{k-1}\|^2$$

$$\leq\left(2\eta^2 M^2(\frac{Q(Q+1)}{2})^2\eta^2\rho^2 + 2\eta^2 L^2(Q+1)^2\right)\|z_k - z_{k-1}\|^2,$$

where $(i)$ follows from the definition of $V_{Qk}$, $(ii)$ follows from Assumption 2 and the fact that $\|I-\eta\nabla_y^2 G(x_k,y_k;\mathcal{B}_j)\| \leq 1$, $(iii)$ follows from Jensen's inequality and $(iv)$ follows because $\|I-\eta\nabla_y^2 G(x_k,y_k;\mathcal{B}_j)\| \leq 1$ and from Assumption 2. Then, the proof is complete. $\qquad\square$

Then, we characterize the Lipschtiz property of $\widehat{\nabla}\Phi(x_k;\mathcal{B}_x)$ defined in eq. (4) in the following lemma.

**Lemma 3.** *Suppose Assumptions 2 holds. Let $\eta < \frac{1}{L}$ and $z = (x, y)$. Then, for $\widehat{\nabla}\Phi(x_k; \mathcal{B}_x)$ defined in eq. (4), we have*

$$\|\widehat{\nabla}\Phi(x_k; \mathcal{B}_x) - \widehat{\nabla}\Phi(x_{k-1}; \mathcal{B}_x)\|^2 \leq L_Q^2 \|z_k - z_{k-1}\|^2, \tag{15}$$

*where $L_Q^2 = 2L^2 + 4\tau^2\eta^2M^2(Q+1)^2 + 8L^4\eta^2(Q+1)^2 + 2L^2\eta^4M^2\rho^2Q^2(Q+1)^2$.*

*Proof.* Based on the form of $\widehat{\nabla}\Phi(x_k; \mathcal{B}_x)$, we have

$$\|\widehat{\nabla}\Phi(x_k; \mathcal{B}_x) - \widehat{\nabla}\Phi(x_{k-1}; \mathcal{B}_x)\|^2$$
$$\leq 2\|\nabla_x F(z_k; \mathcal{B}_F) - \nabla_x F(z_{k-1}; \mathcal{B}_F)\|^2 + 2\|\nabla_x \nabla_y G(x_k, y_k; \mathcal{B}_G)V_{Qk}$$
$$\quad - \nabla_x \nabla_y G(x_{k-1}, y_{k-1}; \mathcal{B}_G)V_{Q(k-1)}\|^2$$
$$\overset{(i)}{\leq} 2L^2\|z_k - z_{k-1}\|^2 + 4\|\nabla_x \nabla_y G(x_k, y_k; \mathcal{B}_G)(V_{Qk} - V_{Q(k-1)})\|^2$$
$$\quad + 4\|(\nabla_x \nabla_y G(x_k, y_k; \mathcal{B}_G) - \nabla_x \nabla_y G(x_{k-1}, y_{k-1}; \mathcal{B}_G))V_{Q(k-1)}\|^2$$
$$\overset{(ii)}{\leq} 2L^2\|z_k - z_{k-1}\|^2 + 4L^2\|V_{Qk} - V_{Q(k-1)}\|^2 + 4\tau^2\|z_k - z_{k-1}\|^2\|V_{Q(k-1)}\|^2$$
$$\overset{(iii)}{\leq} (2L^2 + 4\tau^2\eta^2M^2(Q+1)^2)\|z_k - z_{k-1}\|^2 + 4L^2\|V_{Qk} - V_{Q(k-1)}\|^2$$
$$\overset{(iv)}{\leq} (2L^2 + 4\tau^2\eta^2M^2(Q+1)^2 + 8L^4\eta^2(Q+1)^2 + 2L^2\eta^4M^2\rho^2Q^2(Q+1)^2)\|z_k - z_{k-1}\|^2,$$

where $(i)$ and $(ii)$ follow from Assumption 2, $(iii)$ follows from the fact $\|V_{Qk}\| \leq \eta M(Q+1)$, and $(iv)$ follows from Lemma 1. Then, the proof is complete. $\qquad\square$

**Lemma 4** (**Restatement of Proposition 1**). *Suppose Assumptions 1, 2 and 3 hold. Let $\eta < \frac{1}{L}$. Then, we have*

$$\mathbb{E}\|\widehat{\nabla}\Phi(x_k; \mathcal{B}_x) - \overline{\nabla}\Phi(x_k)\|^2 \leq G^2, \tag{16}$$

*where $G = \frac{2M^2}{S} + \frac{12M^2L^2\eta^2(Q+1)^2}{S} + \frac{4M^2L^2(Q+2)(Q+1)^2\eta^4\sigma^2}{S}$, $\widehat{\nabla}\Phi(x_k; \mathcal{B}_x)$ is defined in eq. (4) and $\overline{\nabla}\Phi(x_k)$ is defined in eq. (3). Further, for the iterative update of line 8 in Algorithm 1, we let $\bar{\epsilon}_k = v_k - \overline{\nabla}\Phi(x_k)$. Then, we have*

$$\mathbb{E}\|\bar{\epsilon}_k\|^2 \leq \mathbb{E}[2\alpha_k^2 G^2 + 2(1 - \alpha_k)^2 L_Q^2\|x_k - x_{k-1}\|^2$$
$$\quad + 2(1 - \alpha_k)^2 L_Q^2\|y_k - y_{k-1}\|^2 + (1 - \alpha_k)^2\|\bar{\epsilon}_{k-1}\|^2], \tag{17}$$

*where $L_Q^2$ is defined in Lemma 3.*

*Proof.* We first prove eq. (16). Based on the forms of $\widehat{\nabla}\Phi(x_k; \mathcal{B}_x)$ and $\overline{\nabla}\Phi(x_k)$, we have

$$\mathbb{E}\|\widehat{\nabla}\Phi(x_k; \mathcal{B}_x) - \overline{\nabla}\Phi(x_k)\|^2$$
$$\overset{(i)}{\leq} 2\mathbb{E}\|\nabla_x F(x_k, y_k; \mathcal{B}_F) - \nabla_x f(x_k, y_k)\|^2$$
$$\quad + 2\mathbb{E}\|\nabla_x \nabla_y G(x_k, y_k; \mathcal{B}_G)V_{Qk} - \nabla_x \nabla_y g(x_k, y_k)\mathbb{E}[V_{Qk}]\|^2$$
$$\overset{(ii)}{\leq} \frac{2M^2}{S} + 2\mathbb{E}\|\nabla_x \nabla_y G(x_k, y_k; \mathcal{B}_G)V_{Qk} - \nabla_x \nabla_y G(x_k, y_k; \mathcal{B}_G)\mathbb{E}[V_{Qk}]$$
$$\quad + \nabla_x \nabla_y G(x_k, y_k; \mathcal{B}_G)\mathbb{E}[V_{Qk}] - \nabla_x \nabla_y g(x_k, y_k)\mathbb{E}[V_{Qk}]\|^2$$
$$\leq \frac{2M^2}{S} + 4\mathbb{E}\|\nabla_x \nabla_y G(x_k, y_k; \mathcal{B}_G)\|^2\mathbb{E}\|V_{Qk} - \mathbb{E}[V_{Qk}]\|^2$$
$$\quad + 4\mathbb{E}\|\nabla_x \nabla_y G(x_k, y_k; \mathcal{B}_G) - \nabla_x \nabla_y g(x_k, y_k)\|^2\|\mathbb{E}[V_{Qk}]\|^2$$
$$\overset{(iii)}{\leq} \frac{2M^2}{S} + 4L^2\mathbb{E}\|V_{Qk} - \mathbb{E}[V_{Qk}]\|^2 + \frac{4L^2}{S}\|\mathbb{E}[V_{Qk}]\|^2$$
$$\overset{(iv)}{\leq} \frac{2M^2}{S} + \frac{12M^2L^2\eta^2(Q+1)^2}{S} + \frac{4M^2L^2(Q+2)(Q+1)^2\eta^4\sigma^2}{S},$$

where $(i)$ follows from the definitions of $\widehat{\nabla}\Phi(x_k;\mathcal{B}_x)$ and $\overline{\nabla}\Phi(x_k)$, $(ii)$ and $(iii)$ follow from Assumption 2, and $(iv)$ follows from Lemma 1 and the bound that

$$\|\mathbb{E}V_{Qk}\|^2 \leq \eta^2 M^2 \|\sum_{q=0}^{Q}(I - \eta\nabla_y^2 g(x_k, y_k))^q\|^2 \leq \eta^2 M^2(Q+1)\sum_{q=0}^{Q}\|(I - \eta\nabla_y^2 g(x_k, y_k))^q\|^2$$
$$\leq \eta^2 M^2(Q+1)^2. \tag{18}$$

Then, we present the proof of eq. (17). Based on the forms of $v_k$ and $\overline{\nabla}\Phi(x_k)$, we have

$$\mathbb{E}\|\bar{\epsilon}_k\| \stackrel{(i)}{=} \mathbb{E}\|\widehat{\nabla}\Phi(x_k;\mathcal{B}_x) + (1-\alpha_k)(v_{k-1} - \widehat{\nabla}\Phi(x_{k-1};\mathcal{B}_x)) - \overline{\nabla}\Phi(x_k)\|^2$$
$$= \mathbb{E}\|\alpha_k(\widehat{\nabla}\Phi(x_k;\mathcal{B}_x) - \overline{\nabla}\Phi(x_k)) + (1-\alpha_k)((\widehat{\nabla}\Phi(x_k;\mathcal{B}_x) - \widehat{\nabla}\Phi(x_{k-1};\mathcal{B}_x))$$
$$- (\overline{\nabla}\Phi(x_k) - \overline{\nabla}\Phi(x_{k-1})) + (1-\alpha_k)(\widehat{\nabla}\Phi(x_{k-1}) - \overline{\nabla}\Phi(x_{k-1}))\|^2$$
$$\stackrel{(ii)}{\leq} \mathbb{E}[2\alpha_k^2\|\widehat{\nabla}\Phi(x_k;\mathcal{B}_x) - \overline{\nabla}\Phi(x_k)\|^2 + 2(1-\alpha_k)^2\|\widehat{\nabla}\Phi(x_k;\mathcal{B}_x) - \widehat{\nabla}\Phi(x_{k-1};\mathcal{B}_x)$$
$$- \overline{\nabla}\Phi(x_k) + \overline{\nabla}\Phi(x_{k-1})\|^2 + (1-\alpha_k)^2\|\bar{\epsilon}_{k-1}\|^2]$$
$$\stackrel{(iii)}{\leq} \mathbb{E}[2\alpha_k^2 G^2 + 2(1-\alpha_k)^2\|\widehat{\nabla}\Phi(x_k;\mathcal{B}_x) - \widehat{\nabla}\Phi(x_{k-1};\mathcal{B}_x)\|^2 + (1-\alpha_k)^2\|\bar{\epsilon}_{k-1}\|^2]$$
$$\stackrel{(iv)}{\leq} \mathbb{E}[2\alpha_k^2 G^2 + 2(1-\alpha_k)^2 L_Q^2\|z_k - z_{k-1}\|^2 + (1-\alpha_k)^2\|\bar{\epsilon}_{k-1}\|^2]$$
$$\stackrel{(v)}{\leq} \mathbb{E}[2\alpha_k^2 G^2 + 2(1-\alpha_k)^2 L_Q^2\|x_k - x_{k-1}\|^2 + 2(1-\alpha_k)^2 L_Q^2\|y_k - y_{k-1}\|^2$$
$$+ (1-\alpha_k)^2\|\bar{\epsilon}_{k-1}\|^2],$$

where $(i)$ follows from the definition of $v_k$, $(ii)$ follows because $\widehat{\nabla}\Phi(x_k;\mathcal{B}_x)$ and $\widehat{\nabla}\Phi(x_k;\mathcal{B}_x) - \widehat{\nabla}\Phi(x_{k-1};\mathcal{B}_x)$ are unbiased estimator of $\overline{\nabla}\Phi(x_k)$ and $\overline{\nabla}\Phi(x_k) - \overline{\nabla}\Phi(x_{k-1})$, respectively, $(iii)$ follows from Lemma 4, $(iv)$ follows from Lemma 3, and $(v)$ follows from the fact that $z_k = (x_k, y_k)$. Then, the proof is complete. $\square$

**Lemma 5.** *Suppose Assumptions 1, 2 and 3 hold. Let $\eta < \frac{1}{L}$. Then, we have*

$$\mathbb{E}\|\nabla_y g(x_k, y_k) - u_k\|^2 \leq \mathbb{E}[2\beta_k^2 G^2 + 2(1-\beta_k)^2 L^2(\|x_k - x_{k-1}\|^2 + \|y_k - y_{k-1}\|^2)$$
$$+ (1-\beta_k)^2\|\nabla_y g(x_{k-1}, y_{k-1}) - u_{k-1}\|^2],$$

*where $G$ is defined in Lemma 4.*

*Proof.* This proof follow from the steps similar to the proof of eq. (17) in Lemma 4. $\square$

Then, we characterize how the variance of the hypergradient and the inner-loop gradient change between iterations.

**Lemma 6.** *Suppose Assumptions 1, 2 and 3 hold. Let $\eta < \frac{1}{L}$, $c_1 \geq \frac{2}{3d^3} + \frac{9\lambda\mu}{4}$, $c_2 \geq \frac{2}{3d^3} + \frac{75L'^2\lambda}{2\mu}$, $\eta_k = \frac{d}{(m+k)^{1/3}}$, $m \geq \max\{2, (c_1 d)^3, (c_2 d)^3, d^3\}$, $\widetilde{x}_{k+1} = x_k - \gamma v_k$, $\widetilde{y}_{k+1} = y_k - \lambda u_k$, where $L'^2 = \max\{(L + \frac{L^2}{\mu} + \frac{M\tau}{\mu} + \frac{LM\rho}{\mu^2})^2, L_Q^2\}$. Then, we have*

$$\frac{1}{\eta_k}\mathbb{E}\|\bar{\epsilon}_{k+1}\|^2 - \frac{1}{\eta_{k-1}}\mathbb{E}\|\bar{\epsilon}_k\|^2 \leq -\frac{9\lambda\mu\eta_k}{4}\mathbb{E}\|\bar{\epsilon}_k\|^2 + 2L_Q^2\eta_k(\|\widetilde{x}_k - x_{k-1}\|^2 + \|\widetilde{y}_k - y_{k-1}\|^2)$$
$$+ \frac{2\alpha_{k+1}^2 G^2}{\eta_k}, \tag{19}$$

*where $L_Q$ is defined in Lemma 3, $G$ and $\bar{\epsilon}_k$ are defined in Lemma 4. Further, we characterize the relationship of the variance of the inner-loop gradient between iterations in the following inequality.*

$$\frac{1}{\eta_k}\mathbb{E}\|\nabla_y g(x_{k+1}, y_{k+1}) - u_{k+1}\|^2 - \frac{1}{\eta_{k-1}}\mathbb{E}\|\nabla_y g(x_k, y_k) - u_k\|^2 \tag{20}$$

$$\leq -\frac{75L'^2\lambda\eta_k}{2\mu}\mathbb{E}\|\nabla_y g(x_k, y_k) - u_k\|^2 + 2L^2\eta_k(\|\widetilde{x}_{k+1} - x_k\|^2 + \|\widetilde{y}_{k+1} - y_k\|^2) + \frac{2\beta_{k+1}^2 G^2}{\eta_k}. \tag{21}$$

*Proof.* We first prove the eq. (19). Based on the forms of $\bar{\epsilon}_k$, we have

$$\frac{1}{\eta_k}\mathbb{E}\|\bar{\epsilon}_{k+1}\|^2 - \frac{1}{\eta_{k-1}}\mathbb{E}\|\bar{\epsilon}_k\|^2$$

$$\overset{(i)}{\leq} \left(\frac{(1-\alpha_{k+1})^2}{\eta_k} - \frac{1}{\eta_{k-1}}\right)\mathbb{E}\|\bar{\epsilon}_k\|^2 + 2(1-\alpha_{k+1})^2 L_Q^2 \eta_k(\|\widetilde{x}_k - x_{k-1}\|^2 + \|\widetilde{y}_k - y_{k-1}\|^2)$$

$$+ \frac{2\alpha_{k+1}^2 G^2}{\eta_k}$$

$$\overset{(ii)}{\leq} \left(\frac{1}{\eta_k} - \frac{1}{\eta_{k-1}} - c_1\eta_k\right)\mathbb{E}\|\bar{\epsilon}_k\|^2 + 2L_Q^2 \eta_k(\|\widetilde{x}_k - x_{k-1}\|^2 + \|\widetilde{y}_k - y_{k-1}\|^2) + \frac{2\alpha_{k+1}^2 G^2}{\eta_k}$$

$$\overset{(iii)}{\leq} -\frac{9\lambda\mu\eta_k}{4}\mathbb{E}\|\bar{\epsilon}_k\|^2 + 2L_Q^2 \eta_k(\|\widetilde{x}_k - x_{k-1}\|^2 + \|\widetilde{y}_k - y_{k-1}\|^2) + \frac{2\alpha_{k+1}^2 G^2}{\eta_k},$$

where $(i)$ follows from eq. (17), $(ii)$ follows because $\alpha_{k+1} = c_1\eta_k^2 \leq c_1\eta_0^2 \leq 1$, and $(iii)$ follows from $c_1 \geq \frac{2}{3d^3} + \frac{9\lambda\mu}{4}$.

Then, we present the proof of eq. (20). In particular, we have

$$\frac{1}{\eta_k}\mathbb{E}\|\nabla_y g(x_{k+1}, y_{k+1}) - u_{k+1}\|^2 - \frac{1}{\eta_{k-1}}\mathbb{E}\|\nabla_y g(x_k, y_k) - u_k\|^2$$

$$\overset{(i)}{\leq} \left(\frac{1}{\eta_k} - \frac{1}{\eta_{k-1}} - c_2\eta_k\right)\mathbb{E}\|\nabla_y g(x_k, y_k) - u_k\|^2 + 2L^2\eta_k(\|\widetilde{x}_{k+1} - x_k\|^2 + \|\widetilde{y}_{k+1} - y_k\|^2)$$

$$+ \frac{2\beta_{k+1}^2 G^2}{\eta_k}$$

$$\overset{(ii)}{\leq} -\frac{75L'^2\lambda\eta_k}{2\mu}\mathbb{E}\|\nabla_y g(x_k, y_k) - u_k\|^2 + 2L^2\eta_k(\|\widetilde{x}_{k+1} - x_k\|^2 + \|\widetilde{y}_{k+1} - y_k\|^2) + \frac{2\beta_{k+1}^2 G^2}{\eta_k},$$

where $(i)$ follows from Lemma 5 and because $\beta_{k+1} = c_2\eta_k^2 \leq c_2\eta_0^2 \leq 1$, and $(ii)$ follows because $c_2 \geq \frac{2}{3d^3} + \frac{75L'^2\lambda}{2\mu}$. Then, the proof is complete. $\qquad\square$

Next, we characterize the approximation bound $C_Q$ on the Hessian inverse.

**Lemma 7.** *Suppose Assumptions 1, 2 and 3 hold. Let $\eta < \frac{1}{L}$. Then, we have*

$$\|\widetilde{\nabla}\Phi(x_k) - \overline{\nabla}\Phi(x_k)\| \leq C_Q,$$

*where $\widetilde{\nabla}\Phi(x_k)$ is defined in eq. (11), $\overline{\nabla}\Phi(x_k)$ is defined in eq. (3), and $C_Q = \frac{(1-\eta\mu)^{Q+1}ML}{\mu}$.*

*Proof.* Following from the proof of Proposition 3 in [20], we have $\|\mathbb{E}[V_{Qk}] - [\nabla_y^2 g(x_k, y_k)]^{-1}\nabla_y f(x_k, y_k)\| \leq \frac{(1-\eta\mu)^{Q+1}M}{\mu}$. Then, we obtain

$$\|\widetilde{\nabla}\Phi(x_k) - \overline{\nabla}\Phi(x_k)\| \leq \|\nabla_x\nabla_y g(x_k, y_k)\|\|\mathbb{E}[V_{Qk}] - [\nabla_y^2 g(x_k, y_k)]^{-1}\nabla_y f(x_k, y_k)\|$$

$$\overset{(i)}{\leq} \frac{(1-\eta\mu)^{Q+1}ML}{\mu},$$

where $(i)$ follows from Assumption 2. Then, the proof is complete. $\qquad\square$

**Lemma 8 (Restatement of Proposition 2).** *Suppose Assumptions 1, 2 and 3 hold. Let $\eta < \frac{1}{L}$, and $\gamma \leq \frac{1}{4L_\Phi\eta_k}$, where $L_\Phi = L + \frac{2L^2+\tau M^2}{\mu} + \frac{\rho LM+L^3+\tau ML}{\mu^2} + \frac{\rho L^2 M}{\mu^3}$. Then, we have*

$$\mathbb{E}[\Phi(x_{k+1})] \leq \mathbb{E}[\Phi(x_k)] + 2\eta_k\gamma L'^2\|y_k - y^*(x_k)\|^2 + 2\eta_k\gamma\|\bar{\epsilon}_k\|^2 + 2\eta_k\gamma C_Q^2 - \frac{1}{2\gamma\eta_k}\|x_{k+1} - x_k\|^2,$$

*where $C_Q$ is defined in Lemma 7 and $L'$ is defined in Proposition 2.*

*Proof.* Based on the Lipschitz property of $\Phi(x_k)$, we have

$$\mathbb{E}[\Phi(x_{k+1})] \overset{(i)}{\leq} \mathbb{E}[\Phi(x_k) + \langle \nabla\Phi(x_k), x_{k+1} - x_k \rangle + \frac{L_\Phi}{2}\|x_{k+1} - x_k\|^2]$$

$$\overset{(ii)}{=} \mathbb{E}[\Phi(x_k) + \eta_k \langle \nabla\Phi(x_k), \widetilde{x}_{k+1} - x_k \rangle + \frac{L_\Phi}{2}\eta_k^2\|\widetilde{x}_{k+1} - x_k\|^2]$$

$$= \mathbb{E}[\Phi(x_k) + \eta_k \langle \nabla\Phi(x_k) - v_k, \widetilde{x}_{k+1} - x_k \rangle + \eta_k \langle v_k, \widetilde{x}_{k+1} - x_k \rangle$$

$$+ \frac{L_\Phi}{2}\eta_k^2\|\widetilde{x}_{k+1} - x_k\|^2],$$

where $(i)$ follows from the smoothness of the function $\Phi(x)$ proved by Lemma 2 in [20], and $(ii)$ follows because $\eta_k(\widetilde{x}_{k+1} - x_k) = x_{k+1} - x_k$, where $\widetilde{x}_{k+1}$ is defined in Lemma 6.

Based on Lemma 25 in [15], we have $\langle v_k, \widetilde{x}_{k+1} - x_k \rangle \leq -\frac{1}{\gamma}\|\widetilde{x}_{k+1} - x_k\|^2$, which yields

$$\langle \nabla\Phi(x_k) - v_k, \widetilde{x}_{k+1} - x_k \rangle$$

$$= \langle \nabla\Phi(x_k) - \widetilde{\nabla}\Phi(x_k) + \widetilde{\nabla}\Phi(x_k) - \overline{\nabla}\Phi(x_k) + \overline{\nabla}\Phi(x_k) - v_k, \widetilde{x}_{k+1} - x_k \rangle$$

$$\leq \|\nabla\Phi(x_k) - \widetilde{\nabla}\Phi(x_k)\|\|\widetilde{x}_{k+1} - x_k\| + \|\widetilde{\nabla}\Phi(x_k) - \overline{\nabla}\Phi(x_k)\|\|\widetilde{x}_{k+1} - x_k\|$$

$$+ \|\overline{\nabla}\Phi(x_k) - v_k\|\|\widetilde{x}_{k+1} - x_k\|$$

$$\overset{(i)}{\leq} 2\gamma L'^2\|y_k - y^*(x_k)\|^2 + \frac{1}{8\gamma}\|\widetilde{x}_{k+1} - x_k\|^2 + C_Q\|\widetilde{x}_{k+1} - x_k\| + 2\gamma\|\overline{\nabla}\Phi(x_k) - v_k\|^2$$

$$+ \frac{1}{8\gamma}\|\widetilde{x}_{k+1} - x_k\|^2$$

$$\overset{(ii)}{\leq} 2\gamma L'^2\|y_k - y^*(x_k)\|^2 + \frac{1}{8\gamma}\|\widetilde{x}_{k+1} - x_k\|^2 + 2\gamma C_Q^2 + \frac{1}{8\gamma}\|\widetilde{x}_{k+1} - x_k\|^2$$

$$+ 2\gamma\|\overline{\nabla}\Phi(x_k) - v_k\|^2 + \frac{1}{8\gamma}\|\widetilde{x}_{k+1} - x_k\|^2,$$

where $(i)$ follows from [20, Lemma 7], Lemma 7 and Young's inequality, and $(ii)$ follows from Young's inequality.

Combining the above inequalities and applying $\gamma \leq \frac{1}{4L_\Phi\eta_k}$, we have

$$\mathbb{E}[\Phi(x_{k+1})] \leq \mathbb{E}[\Phi(x_k)] + 2\eta_k\gamma L'^2\|y_k - y^*(x_k)\|^2 + 2\eta_k\gamma\|\bar{\epsilon}_k\|^2 + 2\eta_k\gamma C_Q^2 - \frac{\eta_k}{2\gamma}\|\widetilde{x}_{k+1} - x_k\|^2$$

$$= \mathbb{E}[\Phi(x_k)] + 2\eta_k\gamma L'^2\|y_k - y^*(x_k)\|^2 + 2\eta_k\gamma\|\bar{\epsilon}_k\|^2 + 2\eta_k\gamma C_Q^2 - \frac{1}{2\gamma\eta_k}\|x_{k+1} - x_k\|^2.$$

Then, the proof is complete. $\qquad\square$

**Lemma 9.** *Suppose Assumptions 1, 2 and 3 hold. Let $\eta_k < 1$ and $0 < \lambda \leq \frac{1}{6L}$. Then, we have*

$$\|y_{k+1} - y^*(x_{k+1})\|^2 \leq \left(1 - \frac{\eta_k\mu\lambda}{4}\right)\|y_k - y^*(x_k)\|^2 - \frac{3\eta_k}{4}\|\widetilde{y}_{k+1} - y_k\|^2$$

$$+ \frac{25\eta_k\lambda}{6\mu}\|\nabla_y g(x_k, y_k) - u_k\|^2 + \frac{25L^2\eta_k}{6\mu^3\lambda}\|x_k - \widetilde{x}_{k+1}\|^2.$$

*Proof.* Based on Lemma 18 in [15] first version, we obtain

$$\|y_{t+1} - y^*(x_{t+1})\|^2 \leq (1 - \frac{\eta_t\tau\lambda}{4})\|y_t - y^*(x_t)\|^2 - \frac{3\eta_t}{4}\|\widetilde{y}_{t+1} - y_t\|^2$$

$$+ \frac{25\eta_t\lambda}{6\tau}\|\nabla_y f(x_t, y_t) - w_t\|^2 + \frac{25\kappa_y^2\eta_t}{6\tau\lambda}\|x_t - \widetilde{x}_{t+1}\|^2,$$

where $\kappa_y = L_f/\tau$. The proof is finished by replacing $f(x_t, y_t)$ with $-g(x_k, y_k)$. $\qquad\square$

## C.2 Proof of Theorem 1

Based on the above lemmas, we develop the proof of Theorem 1 in the following.

**Theorem 3 (Restatement of Theorem 1).** *Apply MRBO to solve the problem in eq.* (1). *Suppose Assumptions 1, 2, and 3 hold. Let the hyperparameters* $c_1 \geq \frac{2}{3d^3} + \frac{9\lambda\mu}{4}, c_2 \geq \frac{2}{3d^3} + \frac{75L'^2\lambda}{2\mu}, m \geq \max\{2, d^3, (c_1d)^3, (c_2d)^3\}, y_1 = y^*(x_1), \eta < \frac{1}{L}, 0 \leq \lambda \leq \frac{1}{6L}, 0 \leq \gamma \leq \min\{\frac{1}{4L_\Phi\eta_K}, \frac{\lambda\mu}{\sqrt{150L'^2L^2/\mu^2 + 8\lambda\mu(L_Q^2 + L^2)}}\}$. *Then, we have*

$$\frac{1}{K}\sum_{k=1}^{K}\left(\frac{L'^2}{4}\|y^*(x_k) - y_k\|^2 + \frac{1}{4}\|\bar{\epsilon}_k\|^2 + \frac{1}{4\gamma^2\eta_k^2}\|x_{k+1} - x_k\|^2\right) \leq \frac{M'}{K}(m+K)^{1/3}, \quad (22)$$

*where* $L'^2$ *is defined in Proposition 2, and* $M' = \frac{\Phi(x_1) - \Phi^*}{\gamma d} + \left(\frac{2G^2(c_1^1 + c_2^2)d^2}{\lambda\mu} + \frac{2C_Q^2 d^2}{\eta_K^2}\right)\log(m + K) + \frac{2G^2}{S\lambda\mu d\eta_0}$.

*Proof.* Firstly, we define a Lyapunov function,

$$\delta_k = \Phi(x_k) + \frac{\gamma}{\lambda\mu}\left(9L'^2\|y_k - y^*(x_k)\|^2 + \frac{1}{\eta_{k-1}}\|\bar{\epsilon}_k\|^2 + \frac{1}{\eta_{k-1}}\|\nabla_y g(x_k, y_k) - u_k\|^2\right).$$

Then, we have

$$\delta_{k+1} - \delta_k$$

$$= \Phi(x_{k+1}) - \Phi(x_k) + \frac{9L'^2\gamma}{\lambda\mu}(\|y_{k+1} - y^*(x_k)\|^2 - \|y_k - y^*(x_k)\|^2)$$

$$+ \frac{\gamma}{\lambda\mu}\left(\frac{1}{\eta_k}\|\bar{\epsilon}_{k+1}\|^2 - \frac{1}{\eta_{k-1}}\|\bar{\epsilon}_k\|^2 + \frac{1}{\eta_k}\|\nabla_y g(x_{k+1}, y_{k+1}) - u_{k+1}\|^2\right.$$

$$\left. - \frac{1}{\eta_{k-1}}\|\nabla_y g(x_k, y_k) - u_k\|^2\right)$$

$$\overset{(i)}{\leq} -\frac{\eta_k}{2\gamma}\|\widetilde{x}_{k+1} - x_k\|^2 + 2\eta_k\gamma L'^2\|y_k - y^*(x_k)\|^2 + 2\eta_k\gamma\|\bar{\epsilon}_k\|^2 + 2\eta_k\gamma C_Q^2$$

$$+ \frac{9L'^2\gamma}{\lambda\mu}\left(-\frac{\eta_k\mu\lambda}{4}\|y_k - y^*(x_k)\|^2 - \frac{3\eta_k}{4}\|\widetilde{y}_{k+1} - y_k\|^2 + \frac{25\eta_k\lambda}{6\mu}\|\nabla_y g(x_k, y_k) - u_k\|^2\right.$$

$$+ \frac{25\kappa_y^2\eta_k}{6\lambda\mu}\|x_k - \widetilde{x}_{k+1}\|^2\left.\right) + \frac{\gamma}{\lambda\mu}\left(-\frac{9\lambda\mu\eta_k}{4}\|\bar{\epsilon}_k\|^2 + 2L_Q^2\eta_k(\|\widetilde{x}_{k+1} - x_k\|^2 + \|\widetilde{y}_{k+1} - y_k\|^2)\right.$$

$$+ \frac{2\alpha_{k+1}^2 G^2}{\eta_k}\left.\right) + \frac{\gamma}{\lambda\mu}\left(-\frac{75L'^2\lambda}{2\mu}\eta_k\|\nabla_y g(x_k, y_k) - u_k\|^2 + 2L^2\eta_k(\|\widetilde{x}_{k+1} - x_k\|^2\right.$$

$$+ \|\widetilde{y}_{k+1} - y_k\|^2) + \frac{2\beta_{k+1}^2 G^2}{\eta_k}\left.\right)$$

$$\overset{(ii)}{\leq} -\frac{L'^2\eta_k\gamma}{4}\|y^*(x_k) - y_k\|^2 - \frac{\gamma\eta_k}{4}\|\bar{\epsilon}_k\|^2 - \frac{\eta_k}{4\gamma}\|\widetilde{x}_{k+1} - x_k\|^2 + \frac{2\alpha_{k+1}^2 G^2\gamma}{\lambda\mu\eta_k} + \frac{2\beta_{k+1}^2 G^2\gamma}{\lambda\mu\eta_k},$$

*where* $(i)$ *follows from Lemmas 6 and 9,* $(ii)$ *follows because* $L' \geq L_Q$ *and* $0 \leq \gamma \leq \frac{\lambda\mu}{\sqrt{150L'^2L^2/\mu^2 + 8\lambda\mu(L_Q^2 + L^2)}}$. *Rearranging the terms in above inequality, we obtain*

$$\frac{L'^2\eta_k}{4}\|y^*(x_k) - y_k\|^2 + \frac{\eta_k}{4}\|\bar{\epsilon}_k\|^2 + \frac{\eta_k}{4\gamma^2}\|\widetilde{x}_{k+1} - x_k\|^2 \leq \frac{\delta_k - \delta_{k+1}}{\gamma} + \frac{2(\alpha_{k+1}^2 + \beta_{k+1}^2)G^2}{\lambda\mu\eta_k}$$

$$+ 2\eta_k C_Q^2. \quad (23)$$

Note that we set $y_1 = y^*(x_1)$ and obtain

$$\delta_1 = \Phi(x_1) + \frac{\gamma}{\lambda\mu}\left(9L'^2\|y_1 - y^*(x_1)\|^2 + \frac{1}{\eta_0}\|\bar{\epsilon}_1\|^2 + \frac{1}{\eta_0}\|\nabla_y g(x_1, y_1) - u_1\|^2\right).$$

Then, telescoping eq. (23) over $k$ from 1 to $K$ yields

$$\frac{1}{K}\sum_{k=1}^{K}\left(\frac{L'^2}{4}\|y^*(x_k)-y_k\|^2+\frac{1}{4}\|\bar{\epsilon}_k\|^2+\frac{1}{4\gamma^2}\|\widetilde{x}_{k+1}-x_k\|^2\right)$$

$$\overset{(i)}{\leq}\frac{1}{K\eta_k\gamma}\left(\Phi(x_1)+\frac{2\gamma G^2}{S\lambda\mu\eta_0}-\Phi^*\right)+\frac{1}{K\eta_k}\sum_{k=1}^{K}\left(\frac{2\alpha_{k+1}^2G^2}{\lambda\mu\eta_k}+\frac{2\beta_{k+1}^2G^2}{\lambda\mu\eta_k}+2\eta_kC_Q^2\right)$$

$$\overset{(ii)}{\leq}\frac{1}{K\eta_k\gamma}(\Phi(x_1)-\Phi^*)+\frac{2G^2}{K\eta_KS\lambda\mu\eta_0}+\frac{(2c_1^2G^2+2c_2G^2)d^3}{K\eta_K\lambda\mu}\log(m+K)$$

$$+\frac{2C_Q^2d^3}{K\eta_K^3}\log(m+K)$$

$$\leq\frac{\Phi(x_1)-\Phi^*}{\gamma d}\frac{(m+K)^{1/3}}{K}+\frac{2G^2}{dS\lambda\mu\eta_0}\frac{(m+K)^{1/3}}{K}$$

$$+\left(\frac{(2c_1^2G^2+2c_2^2G^2)d^2}{\lambda\mu}+\frac{2C_Q^2d^2}{\eta_K^2}\right)\frac{(m+K)^{1/3}}{K}\log(m+K)$$

where $(i)$ follows from eq. (23), $(ii)$ follows because $\sum_{k=1}^{K}\eta_k^3\leq\int_1^K\frac{d^3}{m+k}\leq d^3\log(m+K)$.
We further apply $\|\widetilde{x}_{k+1}-x_k\|=\eta\|x_{k+1}-x_k\|$ to the above inequality and obtain

$$\frac{1}{K}\sum_{k=1}^{K}\left(\frac{L'^2}{4}\|y^*(x_k)-y_k\|^2+\frac{1}{4}\|\bar{\epsilon}_k\|^2+\frac{1}{4\gamma^2\eta_k^2}\|x_{k+1}-x_k\|^2\right)\leq\frac{M'}{K}(m+K)^{1/3}.\quad(24)$$

where $M'=\frac{\Phi(x_1)-\Phi^*}{\gamma d}+\frac{2G^2}{S\lambda\mu d\eta_0}+\left(\frac{2G^2(c_1^1+c_2^2)d^2}{\lambda\mu}+\frac{2C_Q^2d^2}{\eta_K^2}\right)\log(m+K)$. Then, the proof is complete.

$\square$

### C.3 Proof of Corollary 1

**Corollary 3** (**Restatement of Corollary 1**). *Under the same conditions of Theorem 1 and choosing $K=\mathcal{O}(\epsilon^{-1.5}), Q=\mathcal{O}(\log(\frac{1}{\epsilon}))$, MRBO in Algorithm 1 finds an $\epsilon$-stationary point with the gradient complexity of $\mathcal{O}(\epsilon^{-1.5})$ and the (Jacobian-) Hessian-vector complexity of $\widetilde{\mathcal{O}}(\epsilon^{-1.5})$.*

*Proof.* We choose $Q=\mathcal{O}(\log(\frac{1}{\epsilon})), K=\mathcal{O}(\epsilon^{-1.5})$ and $S=\mathcal{O}(1)$, and then have $\mathcal{O}(C_Q)=\mathcal{O}(\epsilon^{-1})$, $M'=\mathcal{O}(1)$, and $m=\mathcal{O}(1)$. Hence, $\mathcal{O}(\frac{M'}{K}(m+K)^{1/3})\leq\mathcal{O}(\frac{M'm^{1/3}}{K}+\frac{M'}{K^{2/3}})=\mathcal{O}(\frac{1}{K^{2/3}})=\mathcal{O}(\epsilon)$, which guarantees the target accuracy. The gradient complexity and Jacobian-vector complexity are given by $KS=\mathcal{O}(\epsilon^{-1.5})$, and the Hessian-vector complexity is given by $KSQ=\widetilde{\mathcal{O}}(\epsilon^{-1.5})$. $\square$

## D  Proof of Theorem 2

### D.1  Proofs of Supporting Lemmas (Propositions 3 and 4 Correspond to Lemmas 18 and 20)

For notation simplification, we define the following:

$$V_{Q\xi}=\eta\sum_{q=-1}^{Q-1}\prod_{j=Q-q}^{Q}(I-\eta\nabla_y^2G(x_k,y_k;\zeta_j))\nabla_yF(x_k,y_k;\xi),$$

which is a single-sample form of $V_{Qk}$ defined in eq. (13). We note that $\|\mathbb{E}[V_{Q\xi}]\|^2=\|\mathbb{E}[V_{Qk}]\|^2\leq\eta^2M^2(Q+1)^2$, where the inequality follows from eq. (18).

Firstly, we characterize the variance of the hypergradients between different iterations.

**Lemma 10.** *Consider Algorithm 2. Suppose Assumptions 2 and 3 hold. Then, we have*

$$\mathbb{E}[\|\widetilde{v}_{k,t} - \overline{\nabla}\Phi(\widetilde{x}_{k,t})\|^2] \leq \mathbb{E}[\|\widetilde{v}_{k,t-1} - \overline{\nabla}\Phi(\widetilde{x}_{k,t-1})\|^2] + \frac{L_Q^2}{S_2}\mathbb{E}[\|\widetilde{x}_{k,t} - \widetilde{x}_{k,t-1}\|^2$$

$$+ \|\widetilde{y}_{k,t} - \widetilde{y}_{k,t-1}\|^2], \tag{25}$$

*where $\overline{\nabla}\Phi(\widetilde{x}_{k,t})$ is defined in eq. (3) and $L_Q$ is defined in Lemma 3.*

*Proof.* In Algorithm 2, the hypergradient estimator $\widetilde{v}_{k,t}$ updates as the following form:

$$\widetilde{v}_{k,t} = \widehat{\nabla}\Phi(\widetilde{x}_{k,t}, \widetilde{y}_{k,t}; \mathcal{S}_2) - \widehat{\nabla}\Phi(\widetilde{x}_{k,t-1}, \widetilde{y}_{k,t-1}; \mathcal{S}_2) + \widetilde{v}_{k,t-1}.$$

Note that

$$\mathbb{E}[\widehat{\nabla}\Phi(\widetilde{x}_{k,t}, \widetilde{y}_{k,t}; \mathcal{S}_2) - \widehat{\nabla}\Phi(\widetilde{x}_{k,t-1}, \widetilde{y}_{k,t-1}; \mathcal{S}_2)|\widetilde{x}_{k,0:t}, \widetilde{y}_{k,0:t}] = \mathbb{E}[\|\widetilde{v}_{k,t} - \widetilde{v}_{k,t-1}\|].$$

Based on Lemma 1 in [4],

$$\mathbb{E}\|\widetilde{v}_{k,t} - \widetilde{v}_{k,t-1} - (\widehat{\nabla}\Phi(\widetilde{x}_{k,t}, \widetilde{y}_{k,t}; \mathcal{S}_2) - \widehat{\nabla}\Phi(\widetilde{x}_{k,t-1}, \widetilde{y}_{k,t-1}; \mathcal{S}_2))\|^2$$

$$\leq \frac{1}{S_2}\mathbb{E}[\|(\widehat{\nabla}\Phi(\widetilde{x}_{k,t}, \widetilde{y}_{k,t}; \xi) - \widehat{\nabla}\Phi(\widetilde{x}_{k,t-1}, \widetilde{y}_{k,t-1}; \xi))\|^2].$$

Furthermore, since $\widehat{\nabla}\Phi(x_k, y_k; \xi)$ is $L_Q$- Lipschitz continuous which is proved in Lemma 3, we have

$$\mathbb{E}[\|\widetilde{v}_{k,t} - \overline{\nabla}\Phi(\widetilde{x}_{k,t})\|^2] \leq \mathbb{E}[\|\widetilde{v}_{k,t-1} - \overline{\nabla}\Phi(\widetilde{x}_{k,t-1})\|^2] + \frac{L_Q^2}{S_2}\mathbb{E}[\|\widetilde{x}_{k,t} - \widetilde{x}_{k,t-1}\|^2$$

$$+ \|\widetilde{y}_{k,t} - \widetilde{y}_{k,t-1}\|^2].$$

Then, the proof is complete. $\qquad\square$

**Lemma 11.** *Suppose Assumptions 2 and 3 hold. Let $\Delta_k = \mathbb{E}\|\overline{\nabla}\Phi(x_k) - v_k\|^2 + \mathbb{E}\|\nabla_y g(x_k, y_k) - u_k\|^2$, and $\widetilde{\Delta}_{k,t} = \mathbb{E}\|\overline{\nabla}\Phi(\widetilde{x}_{k,t}) - \widetilde{v}_{k,t}\|^2 + \mathbb{E}\|\nabla_y g(\widetilde{x}_{k,t}, \widetilde{y}_{k,t}) - \widetilde{u}_{k,t}\|^2$. Then, we have*

$$\widetilde{\Delta}_{k,0} \leq \Delta_k + \frac{2L_Q^2}{S_2}\mathbb{E}(\|x_{k+1} - x_k\|^2),$$

*where $L_Q$ is defined in Lemma 7.*

*Proof.* Based on the form of $\widetilde{\Delta}_{k,0}$, we have

$$\widetilde{\Delta}_{k,0} = \mathbb{E}(\|\widetilde{v}_{k,0} - \overline{\nabla}\Phi(\widetilde{x}_{k,0}, \widetilde{y}_{k,0})\|^2 + \|\widetilde{u}_{k,0} - \nabla_y g(\widetilde{x}_{k,0}, \widetilde{y}_{k,0})\|^2)$$

$$\overset{(i)}{\leq} \mathbb{E}(\|v_k - \overline{\nabla}\Phi(x_k, y_k)\|^2 + \|u_k - \nabla_y g(x_k, y_k)\|^2 + \frac{2L_Q^2}{S_2}\mathbb{E}(\|\widetilde{x}_{k,0} - x_k\|^2 + \|\widetilde{y}_{k,0} - y_k\|^2)$$

$$\overset{(ii)}{=} \Delta_k + \frac{2L_Q^2}{S_2}\mathbb{E}(\|x_{k+1} - x_k\|^2 + \|y_k - y_k\|^2)$$

$$= \Delta_k + \frac{2L_Q^2}{S_2}\mathbb{E}(\|x_{k+1} - x_k\|^2),$$

where $(i)$ follows from Lemma 10, $(ii)$ follows because $\widetilde{y}_{k,0} = y_k$. Then, the proof is complete. $\quad\square$

**Lemma 12.** *Suppose Assumptions 2 and 3 hold. Then, we have*

$$\Delta_k \leq \widetilde{\Delta}_{k-1,0} + \frac{2L_Q^2}{S_2}\beta^2\sum_{t=0}^{m}\|\widetilde{u}_{k-1,t}\|^2,$$

*where $\widetilde{\Delta}_{k-1,0}, \Delta_k$ are defined in Lemma 11 and $L_Q$ is defined in Lemma 7.*

*Proof.* Based on the forms of $\Delta_k$, we have

$$\Delta_k = \mathbb{E}(\|v_k - \overline{\nabla}\Phi(x_k, y_k)\|^2 + \|u_k - \nabla_y g(x_k, y_k)\|^2)$$

$$\overset{(i)}{=} \widetilde{\Delta}_{k-1,m+1}$$

$$\overset{(ii)}{\leq} \mathbb{E}(\|\widetilde{v}_{k-1,0} - \overline{\nabla}\Phi(\widetilde{x}_{k-1,0}, \widetilde{y}_{k-1,0})\|^2 + \|\widetilde{u}_{k-1,0} - \nabla_y g(\widetilde{x}_{k-1,0}, \widetilde{y}_{k-1,0}\|^2)$$

$$+ \frac{2L_Q^2}{S_2} \sum_{t=0}^{m} (\|\widetilde{x}_{k-1,t+1} - \widetilde{x}_{k-1,t}\|^2 + \|\widetilde{y}_{k-1,t+1} - \widetilde{y}_{k-1,t}\|^2)$$

$$\overset{(iii)}{=} \widetilde{\Delta}_{k-1,0} + \frac{2L_Q^2}{S_2}\beta^2 \sum_{t=0}^{m} \|\widetilde{u}_{k-1,t}\|^2,$$

where $(i)$ follows because $u_k = \widetilde{u}_{k-1,m+1}$, and $v_k = \widetilde{v}_{k-1,m+1}$, $(ii)$ follows from Lemma 10, and $(iii)$ follows from the fact that $\widetilde{x}_{k-1,t+1} = \widetilde{x}_{k-1,t}$. Then, the proof is complete. $\qquad\square$

Furtheremore, we characterize the relationship between $\widetilde{u}_{k,t}$ in different iterations.

**Lemma 13.** *Suppose Assumptions 1, 2 and 3 hold. We let $\beta = \frac{2}{13L_Q}$, and $S_2 \geq 2(\frac{L}{\mu} + 1)L\beta$. Then, we have*

$$\mathbb{E}[\|\widetilde{u}_{k,t}\|^2 | \mathcal{F}_{k,t}] \leq a\|\widetilde{u}_{k,t-1}\|^2, \tag{26}$$

*where $a = \left(1 - \frac{\beta\mu L}{\mu + L}\right)$, $\widetilde{u}_{k,t}$ is defined in Algorithm 2, and $\mathcal{F}_{k,t}$ denotes all information of $\{\widetilde{y}_{k,j}\}_{j=0}^{t}$ and $\{\widetilde{u}_{k,j}\}_{j=0}^{t-1}$.*

*Proof.* Based on the definition of $\widetilde{u}_{k,t}$, we have

$$\mathbb{E}[\|\widetilde{u}_{k,t}\|^2 | \mathcal{F}_{k,t}]$$

$$= \|\widetilde{u}_{k,t-1}\|^2 + 2\mathbb{E}[\langle \widetilde{u}_{k,t-1}, \nabla_y G(\widetilde{y}_{k,t}) - \nabla_y G(\widetilde{y}_{k,t-1})\rangle | \mathcal{F}_{k,t}]$$

$$+ \mathbb{E}[\|\nabla_y G(\widetilde{y}_{k,t}) - \nabla_y G(\widetilde{y}_{k,t-1})\|^2 | \mathcal{F}_{k,t}]$$

$$= \|\widetilde{u}_{k,t-1}\|^2 - \frac{\beta}{2}\mathbb{E}[\langle \widetilde{y}_{k,t} - \widetilde{y}_{k,t-1}, \nabla_y g(\widetilde{y}_{k,t}) - \nabla_y g(\widetilde{y}_{k,t-1})\rangle]$$

$$+ [\|\nabla_y G(\widetilde{y}_{k,t}) - \nabla_y G(\widetilde{y}_{k,t-1})\|^2 | \mathcal{F}_{k,t}]$$

$$\overset{(i)}{\leq} \|\widetilde{u}_{k,t-1}\|^2 - \frac{2}{\beta}\left(\frac{\mu L}{\mu + L}\|\widetilde{y}_{k,t} - \widetilde{y}_{k,t-1}\|^2 + \frac{1}{\mu + L}\|\nabla_y g(\widetilde{y}_{k,t} - \nabla_y g(\widetilde{y}_{k,t-1})\|^2\right)$$

$$+ \mathbb{E}[\|\nabla_y G(\widetilde{y}_{k,t}) - \nabla_y G(\widetilde{y}_{k,t-1})\|^2 | \mathcal{F}_{k,t}]$$

$$\leq \left(1 - \frac{2\beta\mu L}{\mu + L}\right)\|\widetilde{u}_{k,t-1}\|^2 - \left(\frac{2}{\beta(\mu + L)} - 2\right)\|\nabla_y g(\widetilde{y}_{k,t}) - \nabla_y g(\widetilde{y}_{k,t-1})\|^2]$$

$$+ 2\mathbb{E}[\|\nabla_y G(\widetilde{y}_{k,t}) - \nabla_y G(\widetilde{y}_{k,t-1}) - [\nabla_y g(\widetilde{y}_{k,t}) - \nabla_y g(\widetilde{y}_{k,t-1})]\|^2 | \mathcal{F}_{k,t}]$$

$$\overset{(ii)}{\leq} \left(1 - \frac{2\beta\mu L}{\mu + L}\right)\|\widetilde{u}_{k,t-1}\|^2 + 2\mathbb{E}[\|\nabla_y G(\widetilde{y}_{k,t}) - \nabla_y G(\widetilde{y}_{k,t-1}) - (\nabla_y g(\widetilde{y}_{k,t}) - \nabla_y g(\widetilde{y}_{k,t-1}))\|^2]$$

$$\overset{(iii)}{\leq} \left(1 - \frac{2\beta\mu L}{\mu + L}\right)\|\widetilde{u}_{k,t-1}\|^2 + \frac{2L^2}{S_2}\|\widetilde{y}_{k,t} - \widetilde{y}_{k,t-1}\|^2$$

$$= \left(1 - \frac{2\beta\mu L}{\mu + L} + \frac{2L^2\beta^2}{S_2}\right)\|\widetilde{u}_{k,t-1}\|^2$$

$$\overset{(iv)}{\leq} \left(1 - \frac{\beta\mu L}{\mu + L}\right)\|\widetilde{u}_{k,t-1}\|^2$$

where $(i)$ follows from Assumptions 2 and 3, $(ii)$ follows from the fact that $\beta \leq \frac{1}{2L}$, $(iii)$ follows from Lemma 10, and $(iv)$ follows because $S_2 \geq 2(\frac{L}{\mu} + 1)L\beta$. Then, the proof is complete. $\qquad\square$

Furthermore, we characterize the relationship among $u_k$, $\delta_k$ and $\Delta_k$.

**Lemma 14.** *Suppose Assumptions 2 and 3 hold. Let* $\delta_k = \mathbb{E}\|\nabla_y g(x_k, y_k)\|^2$, $\beta = \frac{2}{13L_Q}$, *and* $S_2 \geq 2(\frac{L}{\mu}+1)L\beta$. *Then, we have*

$$\Delta_k \leq \widetilde{\Delta}_{k-1,0} + \frac{2L_Q^2\beta^2}{S_2(1-a)}\mathbb{E}\|\widetilde{u}_{k-1,0}\|^2,$$

*where* $\widetilde{\Delta}_{k-1,0}$ *and* $\Delta_k$ *are defined in Lemma 11, and* $a$ *is defined in Lemma 13, and*

$$\mathbb{E}\|\widetilde{u}_{k,0}\|^2 \leq 3(\widetilde{\Delta}_{k,0} + \mathbb{E}\|\nabla_y g(x_{k+1}, y_k) - \nabla_y g(x_k, y_k)\|^2 + \delta_k).$$

*Proof.* Based on eq.(23) and eq.(24) in [27] first version, we obtain:

$$\delta_k \leq \widetilde{\delta}_{k-1,0} + \frac{l^2\lambda^2}{S_2(1-\alpha)}\|\widetilde{u}_{k-1,0}\|^2$$

and

$$\mathbb{E}\|\widetilde{u}_{k,0}\|^2 \leq 3(\widetilde{\Delta}_{k,0} + \mathbb{E}\|\nabla_y f(x_{k+1}, y_k) - \nabla_y f(x_k, y_k)\|^2 + \delta_k).$$

The proof is finished by replacing $l$ with $L_Q$ and replacing $f(x, y)$ with $g(x, y)$. $\qquad\square$

Next, we characterize the recursive updates of $\delta_k$ and $\Delta_k$, respectively.

**Lemma 15.** *Suppose Assumptions 1, 2 and 3 hold. Let* $\beta = \frac{2}{13L_Q}$, *and* $S_2 \geq 2(\frac{L}{\mu}+1)L\beta$. *Then, we have*

$$\delta_{k+1} \leq \frac{4}{\mu\beta(m+1)}(\mathbb{E}\|\nabla_y g(x_{k+1}, y_k) - \nabla_y g(x_k, y_k)\|^2 + \delta_k) + \frac{L\beta}{2-L\beta}\mathbb{E}\|\widetilde{u}_{k,0}\|^2 + \widetilde{\Delta}_{k,0}.$$

*where* $\delta_k$ *is defined in Lemma 14 and* $\widetilde{\Delta}_{k,0}$ *is defined in Lemma 11.*

*Proof.* Following from Lemma 12 in [27], we have

$$\mathbb{E}\|\nabla_y g(x_{k+1}, \widetilde{y}_{k,t+1})\|^2 \leq \frac{2}{\mu\beta(m+1)}\|\nabla_y g(x_{k+1}, \widetilde{y}_{k,0})\|^2 + \frac{L\beta}{2-L\beta}\mathbb{E}\|\widetilde{u}_{k,0}\|^2$$
$$+ \mathbb{E}\|\nabla_y g(x_{k+1}, \widetilde{y}_{k,0}) - \widetilde{u}_{k,0}\|^2.$$

Since $\widetilde{y}_{k,0} = y_k, x_{k+1} = \widetilde{x}_{k,0}, \widetilde{y}_{k,m+1} = y_{k+1}, \delta_{k+1} = \mathbb{E}\|\nabla_y g(x_{k+1}, y_{k+1})\|^2$, we have

$$\delta_{k+1} \leq \frac{2}{\mu\beta(m+1)}\|\nabla_y g(x_{k+1}, y_k)\|^2 + \frac{L\beta}{2-L\beta}\mathbb{E}\|\widetilde{u}_{k,0}\|^2 + \mathbb{E}\|\nabla_y g(x_{k+1}, y_k) - \widetilde{u}_{k,0}\|^2$$
$$\leq \frac{2}{\mu\beta(m+1)}\mathbb{E}\|\nabla_y g(x_{k+1}, y_k) - \nabla_y g(x_k, y_k) + \nabla_y g(x_k, y_k)\|^2 + \frac{L\beta}{2-L\beta}\mathbb{E}\|\widetilde{u}_{k,0}\|^2$$
$$+ \mathbb{E}\|\nabla_y g(\widetilde{x}_{k,0}, \widetilde{y}_{k,0}) - \widetilde{u}_{k,0}\|^2$$
$$\leq \frac{4}{\mu\beta(m+1)}(\mathbb{E}\|\nabla_y g(x_{k+1}, y_k) - \nabla_y g(x_k, y_k)\|^2 + \delta_k) + \frac{L\beta}{2-L\beta}\mathbb{E}\|\widetilde{u}_{k,0}\|^2 + \widetilde{\Delta}_{k,0}.$$

Then, the proof is complete. $\qquad\square$

**Lemma 16.** *Suppose Assumptions 2 and 3 hold. Let* $\beta = \frac{2}{13L_Q}$, *and* $S_2 \geq 2(\frac{L}{\mu}+1)L\beta$. *Then, we have*

$$\Delta_k \leq \frac{\alpha^2 L_Q^2}{S_2}\left(2 + \frac{12L_Q^2\beta^2}{S_2(1-a)} + \frac{6L^2\beta^2}{S_2(1-a)}\|v_{k-1}\|^2 + \frac{6L_Q^2\beta^2}{S_2(1-a)}\delta_{k-1}\right)$$
$$+ \left(1 + \frac{6L_Q^2\beta^2}{S_2(1-a)}\right)\Delta_{k-1},$$

*where* $\delta_{k-1}$ *is defined in Lemma 14,* $\Delta_k$ *is defined in Lemma 11 and* $L_Q$ *is defined in Lemma 3.*

*Proof.* Based on the bounds on $\Delta_k$ in Lemma 14, we have

$$\Delta_k \leq \widetilde{\Delta}_{k-1,0} + \frac{2L_Q^2\beta^2}{S_2(1-a)}\mathbb{E}\|\widetilde{u}_{k-1,0}\|^2$$

$$\overset{(i)}{\leq} \widetilde{\Delta}_{k-1,0} + \frac{6L_Q^2\beta^2}{S_2(1-a)}(\widetilde{\Delta}_{k-1,0} + \|\nabla_y g(x_k, y_{k-1}) - \nabla_y g(x_{k-1}, y_{k-1})\|^2 + \delta_{k-1})$$

$$\overset{(ii)}{\leq} \left(1 + \frac{6L_Q^2\beta^2}{S_2(1-a)}\right)\widetilde{\Delta}_{k-1,0} + \frac{6L_Q^2\beta^2}{S_2(1-a)}(L^2\alpha^2\|v_{k-1}\|^2 + \delta_{k-1})$$

$$\overset{(iii)}{\leq} \left(1 + \frac{6L_Q^2\beta^2}{S_2(1-a)}\right)\left(\Delta_{k-1} + \frac{2L_Q^2\alpha^2}{S_2}\|v_{k-1}\|^2\right) + \frac{6L_Q^2\beta^2}{S_2(1-a)}(L^2\alpha^2\|v_{k-1}\|^2 + \delta_{k-1})$$

$$\leq \frac{\alpha^2 L_Q^2}{S_2}\left(2 + \frac{12L_Q^2\beta^2}{S_2(1-a)} + \frac{6L^2\beta^2}{S_2(1-a)}\|v_{k-1}\|^2 + \frac{6L_Q^2\beta^2}{S_2(1-a)}\delta_{k-1}\right)$$

$$+ \left(1 + \frac{6L_Q^2\beta^2}{S_2(1-a)}\right)\Delta_{k-1},$$

where $(i)$ follows from Lemma 14, $(ii)$ follows from Assumption 2, and $(iii)$ follows from Lemma 11. Then, the proof is complete. $\qquad\square$

**Lemma 17.** *Suppose Assumptions 1, 2 and 3 hold. Let $\beta = \frac{2}{13L_Q}$, and $S_2 \geq 2(\frac{L}{\mu}+1)L\beta$. Then, we have*

$$\delta_k \leq \left(\frac{4L^2\alpha^2}{\mu\beta(m+1)} + \frac{3L^3\beta\alpha^2}{2-L\beta} + \frac{6LL_Q^2\alpha^2\beta}{2-L\beta} + 2L_Q^2\alpha^2\right)\mathbb{E}\|v_{k-1}\|^2$$

$$+ \frac{2+2L\beta}{2-L\beta}\Delta_{k-1} + \left(\frac{4}{\mu\beta(m+1)} + \frac{3L\beta}{2-L\beta}\right)\delta_{k-1},$$

*where $\delta_k$ is defined in Lemma 14, $\Delta_{k-1}$ is defined in Lemma 11 and $L_Q$ is defined in Lemma 3.*

*Proof.* Based on the bounds of $\delta_k$ in Lemma 15, we have

$$\delta_k \leq \frac{4}{\mu\beta(m+1)}(L^2\alpha^2\|v_{k-1}\|^2 + \delta_{k-1}) + \frac{L\beta}{2-L\beta}\mathbb{E}\|\widetilde{u}_{k-1,0}\|^2 + \widetilde{\Delta}_{k-1,0}$$

$$\overset{(i)}{\leq} \frac{4}{\mu\beta(m+1)}(L^2\alpha^2\|v_{k-1}\|^2 + \delta_{k-1}) + \frac{3L\beta}{2-L\beta}(\widetilde{\Delta}_{k-1,0} + L^2\alpha^2\|v_{k-1}\|^2 + \delta_{k-1}) + \widetilde{\Delta}_{k-1,0}$$

$$= \left(\frac{4}{\mu\beta(m+1)} + \frac{3L\beta}{2-L\beta}\right)\delta_{k-1} + \left(\frac{4L^2\alpha^2}{\mu\beta(m+1)} + \frac{3L^2\beta\alpha^2}{2-L\beta}\right)\|v_{k-1}\|^2$$

$$+ \left(1 + \frac{3L\beta}{2-L\beta}\right)\widetilde{\Delta}_{k-1,0}$$

$$\overset{(ii)}{\leq} \left(\frac{4}{\mu\beta(m+1)} + \frac{3L\beta}{2-L\beta}\right)\delta_{k-1} + \left(1 + \frac{3L\beta}{2-L\beta}\right)\left(\Delta_{k-1} + \frac{2L_Q^2\alpha^2}{S_2}\|v_{k-1}\|^2\right)$$

$$+ \left(\frac{4L^2\alpha^2}{\mu\beta(m+1)} + \frac{3L^3\beta\alpha^2}{2-L\beta}\right)\|v_{k-1}\|^2$$

$$\leq \left(\frac{4L^2\alpha^2}{\mu\beta(m+1)} + \frac{3L^3\beta\alpha^2}{2-L\beta} + \frac{6LL_Q^2\alpha^2\beta}{2-L\beta} + 2L_Q^2\alpha^2\right)\mathbb{E}\|v_{k-1}\|^2$$

$$+ \frac{2+2L\beta}{2-L\beta}\Delta_{k-1} + \left(\frac{4}{\mu\beta(m+1)} + \frac{3L\beta}{2-L\beta}\right)\delta_{k-1},$$

where $(i)$ follows from Lemma 14, and $(ii)$ follows from Lemma 11. Then, the proof is complete. $\quad\square$

**Lemma 18 (Restatement of Proposition 3).** *Suppose Assumptions 1, 2 and 3 hold. Let $\eta < \frac{1}{L}$. Then, we have*

$$\mathbb{E}\|\widehat{\nabla}\Phi(x_k, y_k; \xi) - \overline{\nabla}\Phi(x_k)\|^2 \leq \sigma'^2, \tag{27}$$

where $\sigma'^2 = 2M^2 + 28(Q+1)^2 L^2 M^2 \eta^2$, $\widehat{\nabla}\Phi(x_k, y_k; \xi)$ is defined in eq. (5) with single sample $\xi$, and $\overline{\nabla}\Phi(x_k)$ is defined in eq. (3). Furthermore, let $\beta = \frac{2}{13L_Q}$, $q = (1-a)S_2$, and $m = \frac{16}{\mu\beta} - 1$. Then, we have

$$\sum_{k=0}^{K-1} \Delta_k \le \frac{4\sigma'^2 K}{S_1} + 22\alpha^2 L_Q^2 \sum_{k=0}^{K-2} \mathbb{E}\|v_k\|^2 + \frac{4}{3}\delta_0, \tag{28}$$

where $\delta_0$ is defined in Lemma 14.

*Proof.* We first prove eq. (27). Based on the forms of $\widehat{\nabla}\Phi(x_k, y_k; \xi)$ and $\overline{\nabla}\Phi(x_k)$, we have

$\mathbb{E}\|\widehat{\nabla}\Phi(x_k, y_k; \xi) - \overline{\nabla}\Phi(x_k)\|^2$

$\le \mathbb{E}\|\nabla_x F(x_k, y_k; \xi) - \nabla_x f(x_k, y_k) - (\nabla_x\nabla_y G(x_k, y_k; \zeta)V_{Q\xi} - \nabla_x\nabla_y g(x_k, y_k)\mathbb{E}[V_{Q\xi}])\|^2$

$\overset{(i)}{\le} 2M^2 + 4\mathbb{E}\|\nabla_x\nabla_y G(x_k, y_k; \zeta)V_{Q\xi} - (\nabla_x\nabla_y G(x_k, y_k; \zeta)\mathbb{E}[V_{Q\xi}])\|^2$

$\qquad + 4\mathbb{E}\|(\nabla_x\nabla_y G(x_k, y_k; \zeta)\mathbb{E}[V_{Q\xi}]) - \nabla_x\nabla_y g(x_k, y_k)\mathbb{E}[V_{Q\xi}])\|^2$

$\overset{(ii)}{\le} 2M^2 + 4L^2\mathbb{E}\|V_{Q\xi} - \mathbb{E}[V_{Q\xi}]\|^2 + 4\|\mathbb{E}[V_{Q\xi}]\|^2\mathbb{E}\|\nabla_x\nabla_y G(x_k, y_k; \zeta) - \nabla_x\nabla_y g(x_k, y_k)\|^2$

$\overset{(iii)}{\le} 2M^2 + 8L^2\mathbb{E}\|\eta\sum_{q=-1}^{Q-1}\prod_{j=Q-q}^{Q}(I - \eta\nabla_y^2 G(x_k, y_k; \zeta_j))\nabla_y F(x_k, y_k; \xi)$

$\qquad - \eta\sum_{q=-1}^{Q-1}\prod_{j=Q-q}^{Q}(I - \eta\nabla_y^2 G(x_k, y_k; \zeta_j))\nabla_y f(x_k, y_k)\|^2$

$\qquad + 8L^2\mathbb{E}\|\eta\sum_{q=-1}^{Q-1}\prod_{j=Q-q}^{Q}(I - \eta\nabla_y^2 G(x_k, y_k; \zeta_j))\nabla_y f(x_k, y_k)$

$\qquad - \eta\sum_{q=-1}^{Q-1}\prod_{j=Q-q}^{Q}(I - \eta\nabla_y^2 g(x_k, y_k))\nabla_y f(x_k, y_k)\|^2 + 4\eta^2 M^2 L^2(Q+1)^2$

$\overset{(iv)}{\le} 2M^2 + 8L^2\eta^2(Q+1)^2 M^2 + 16L^2\eta^2 M^2(Q+1)^2 + 4\eta^2 M^2 L^2(Q+1)^2$

$= 2M^2 + 28(Q+1)^2 L^2 M^2 \eta^2 = \sigma'^2,$

where $(i)$ and $(ii)$ follows from Assumption 2, $(iii)$ follows because $\|\mathbb{E}[V_{Q\xi}]\|^2 = \|\mathbb{E}[V_{Qk}]\|^2 \le \eta^2 M^2(Q+1)^2$ in eq. (18), and $(iv)$ follows because $\|(I - \eta\nabla_y^2 G(x_k, y_k; \zeta))\| \le 1$.

Then, we present the proof of eq. (28). Based on the bound on $\Delta_k$ in Lemma 16, we have

$$\Delta_k \le \left(1 + \frac{6L_Q^2\beta^2}{S_2(1-a)}\right)\Delta_{k-1} + \frac{\alpha^2 L_Q^2}{S_2}\left(2 + \frac{12L_Q^2\beta^2}{1-a} + \frac{6L^2\beta^2}{1-a}\right)\|v_{k-1}\|^2$$

$$+ \frac{6L_Q^2\beta^2}{S_2(1-a)}\delta_{k-1}$$

$$\le \frac{\alpha^2 L_Q^2}{S_2}\left(2 + \frac{12\beta^2(L^2 + L_Q^2)}{1-a}\right)\sum_{p=k'}^{K-1}\left(1 + \frac{6L_Q^2}{S_2(1-a)}\right)^{p-k'}\mathbb{E}\|v_{K-1+k'-p}\|^2$$

$$+ \frac{6L_Q^2\beta^2}{S_2(1-a)}\sum_{p=k'}^{K-1}\left(1 + \frac{6L_Q^2\beta^2}{S_2(1-a)}\right)^{p-k'}\delta_{K-1+k'-p} + \left(1 + \frac{6L_Q^2\beta^2}{S_2(1-a)}\right)^{k-k'}\Delta_{k'}$$

$$\overset{(i)}{\le} \frac{3}{2}\Delta_{k'} + \frac{3\alpha^2 L_Q^2}{S_2}\left(1 + \frac{6\beta(L^2 + L_Q^2)}{1-a}\right)\sum_{p=k'}^{K-1}\mathbb{E}\|v_{K-1+k'-p}\|^2$$

$$+ \frac{9L_Q^2\beta^2}{S_2(1-a)}\sum_{p=k'}^{K-1}\delta_{K-1+k'-p},$$

where $(i)$ follows from the following bound:

$$\left(1 + \frac{L_Q^2\beta^2}{S_1(1-a)}\right)^{p-k'} \le \left(1 + \frac{6L_Q^2\beta^2}{S_2(1-a)}\right)^q \le 1 + \frac{\frac{6L_Q^2\beta^2 q}{S_2(1-a)}}{1 - \frac{6L_Q^2\beta^2(q-1)}{S_2(1-a)}}$$

$$\leq 1 + \frac{6L_Q^2\beta^2}{1 - \frac{6L_Q^2\beta^2 q}{S_2(1-a)}} < \frac{3}{2}$$

where $\beta = \frac{2}{13L_Q}$, and $q = (1-a)S_2$. Then telescoping $\Delta_k$ over $k$ from $(n_k - 1)q$ to $K - 1$, we have

$$\sum_{k=(n_k-1)q}^{K-1} \Delta_k \leq \frac{3\alpha L_Q^2}{S_2}\left(1 + \frac{6\beta^2(L^2 + L_Q^2)}{1-a}\right) \sum_{k=(n_k-1)q}^{K-1}\sum_{p=k'}^{k} \mathbb{E}\|v_{K-1+k'-p}\|^2$$

$$+ \frac{9L_Q^2\beta^2}{S_2(1-a)} \sum_{k=(n_k-1)q}^{K-1}\sum_{p=k'}^{k-1} \delta_{K-1+k'-p} + \frac{3}{2}(K - (n_k-1)q)\Delta_{(n_k-1)q}.$$

Since

$$\sum_{k=(n_k-1)q}^{K-1}\sum_{p=k'}^{k} \mathbb{E}\|v_{K-1+k'-p}\|^2 \leq q \sum_{k=(n_k-1)q}^{K-2} \mathbb{E}\|v_k\|^2,$$

and

$$\sum_{k=(n_k-1)q}^{K-1}\sum_{p=k'}^{k-1} \delta_{K-1+k'-p} \leq q \sum_{k=(n_k-1)q}^{K-2} \delta_k,$$

we have

$$\sum_{k=(n_k-1)q}^{K-1} \Delta_k \leq \frac{3}{2}(K - (n_k-1)q)\Delta_{(n_k-1)q} + \frac{3\alpha^2 L_Q^2 q}{S_2}\left(1 + \frac{6\beta^2(L^2 + L_Q^2)}{1-a}\right) \sum_{k=(n_k-1)q}^{K-2} \mathbb{E}\|v_k\|^2$$

$$+ \frac{9L_Q^2\beta^2 q}{S_2(1-a)} \sum_{k=(n_k-1)q}^{K-2} \delta_k.$$

Futhermore, we assume that $\sigma' \geq \sigma$ and derive the following bound on the initial update in each epoch:

$$\sum_{k=(n_K-n_k)q}^{(n_K-n_k+1)q-1} \Delta_k \leq \frac{3\sigma'^2 q}{2S_1} + \frac{3\alpha^2 L_Q^2 q}{S_2}\left(1 + \frac{6\beta^2(L^2 + L_Q^2)}{1-a}\right) \sum_{k=(n_K-n_k)q}^{(n_K-n_k+1)q-1} \mathbb{E}\|v_k\|^2$$

$$+ \frac{9L_Q^2\beta^2 q}{S_2(1-a)} \sum_{k=(n_K-n_k)q}^{(n_K-n_k+1)q-1} \delta_k.$$

Based on the above inequality, we telescope $\Delta_k$ over $k$ from $0$ to $K - 1$, and obtain

$$\sum_{k=0}^{K-1} \Delta_k \leq \frac{3\sigma'^2 K}{2S_1} + \frac{3\alpha^2 L_Q^2 q}{S_2}\left(1 + \frac{6\beta^2(L^2 + L_Q^2)}{1-a}\right) \sum_{k=0}^{K-2} \mathbb{E}\|v_k\|^2 + \frac{9L_Q^2\beta^2 q}{S_2(1-a)} \sum_{k=0}^{K-2} \delta_k$$

$$\stackrel{(i)}{\leq} \frac{3\sigma'^2 K}{2S_1} + 6\alpha^2 L_Q^2 \sum_{k=0}^{K-2} \mathbb{E}\|v_k\|^2 + \frac{1}{4}\sum_{k=0}^{K-2} \delta_k,$$

where $(i)$ follows because $\beta = \frac{2}{13L_Q}$, and $q = (1-a)S_2$. We further derive the following bound on $\delta_k$:

$$\delta_k \leq \left(\frac{4L^2\alpha^2}{\mu\beta(m+1)} + \frac{3L^3\beta\alpha^2}{2 - L\beta} + \frac{6LL_Q^2\alpha^2\beta}{2 - L\beta} + 2L_Q^2\alpha^2\right)\mathbb{E}\|v_{k-1}\|^2$$

$$+ \frac{2 + 2L\beta}{2 - L\beta}\Delta_{k-1} + \left(\frac{4}{\mu\beta(m+1)} + \frac{3L\beta}{2 - L\beta}\right)\delta_{k-1}$$

$$\overset{(ii)}{\leq} \frac{1}{2}\delta_{k-1} + \frac{13}{4}L_Q^2\alpha^2\mathbb{E}\|v_{k-1}\|^2 + \frac{5}{4}\Delta_{k-1},$$

where $(ii)$ follows because $\beta = \frac{2}{13L_Q}$, $q = (1-a)S_2$, and $m = \frac{16}{\mu\beta} - 1$. Then, we telescope $\delta_k$ and $\Delta_k$ over $k = 0$ to $K - 1$, and have

$$\sum_{k=0}^{K-1} \delta_k \leq 2\delta_0 + \frac{13}{2}L_Q^2\alpha^2 \sum_{k=0}^{K-2} \mathbb{E}\|v_k\|^2 + \frac{5}{2}\sum_{k=0}^{K-2}\Delta_k, \tag{29}$$

and

$$\sum_{k=0}^{K-1}\Delta_k \leq \frac{3\sigma'^2 K}{2S_1} + 6\alpha^2 L_Q^2 \sum_{k=0}^{K-2}\mathbb{E}\|v_k\|^2 + \frac{1}{2}\delta_0 + \frac{13}{8}L_Q^2\alpha^2\sum_{k=0}^{K-3}\mathbb{E}\|v_k\|^2 + \frac{5}{8}\sum_{k=0}^{K-2}\Delta_k$$

$$\leq \frac{3\sigma'^2 K}{2S_1} + 8\alpha^2 L_Q^2\sum_{k=0}^{K-2}\mathbb{E}\|v_k\|^2 + \frac{1}{2}\delta_0 + \frac{5}{8}\sum_{k=0}^{K-2}\Delta_k.$$

Finally, we rearrange the terms in the above bound and obtain

$$\sum_{k=0}^{K-1}\Delta_k \leq \frac{4\sigma'^2 K}{S_1} + 22\alpha^2 L_Q^2\sum_{k=0}^{K-2}\mathbb{E}\|v_k\|^2 + \frac{4}{3}\delta_0,$$

Then, the proof is complete. $\qquad\square$

**Lemma 19.** *Suppose Assumptions 1, 2 and 3 hold. Let $\beta = \frac{2}{13L_Q}$, $q = (1-a)S_2$, and $m = \frac{16}{\mu\beta} - 1$. Then, we have*

$$\sum_{k=0}^{K-1}\delta_k \leq \frac{10\sigma'^2 K}{S_1} + 6\delta_0 + 62\alpha^2 L_Q^2\sum_{k=0}^{K-2}\mathbb{E}\|v_k\|^2,$$

*where $L_Q$ is defined in Lemma 3 and $\sigma'$ is defined in Lemma 18.*

*Proof.* Based on the inequalities in eq. (29) and eq. (28), we have

$$\sum_{k=0}^{K-1}\delta_k \leq \frac{10\sigma'^2 K}{S_1} + 55\alpha^2 L_Q^2\sum_{k=0}^{K-2}\mathbb{E}\|v_k\|^2 + \frac{10}{3}\delta_0 + 2\delta_0 + \frac{13}{2}L_Q^2\alpha^2\sum_{k=0}^{K-2}\mathbb{E}\|v_k\|^2$$

$$\leq \frac{10\sigma'^2 K}{S_1} + 6\delta_0 + 62\alpha^2 L_Q^2\sum_{k=0}^{K-2}\mathbb{E}\|v_k\|^2.$$

Then, the proof is complete. $\qquad\square$

**Lemma 20** (**Restatement of Proposition 4**). *Suppose Assumptions 1,2 and 3 hold. Then, we have*

$$\mathbb{E}[\Phi(x_{k+1})] \leq \mathbb{E}[\Phi(x_k)] + \frac{\alpha L'^2}{\mu^2}\mathbb{E}\|\nabla_y g(x_k, y_k)\|^2 + \alpha\mathbb{E}\|\widetilde{\nabla}\Phi(x_k) - v_k\|^2$$

$$- \left(\frac{\alpha}{2} - \frac{\alpha^2}{2}L_\Phi\right)\mathbb{E}\|v_k\|^2. \tag{30}$$

*where $L' = L + \frac{L^2}{\mu} + \frac{M\tau}{\mu} + \frac{LM\rho}{\mu^2}$, and $\widetilde{\nabla}\Phi(x_k)$ is defined in eq. (11).*

*Proof.* Based on the smoothness of the function $\Phi(x)$, we have

$$\Phi(x_{k+1}) \overset{(i)}{\leq} \Phi(x_k) + \langle\nabla\Phi(x_k), x_{k+1} - x_k\rangle + \frac{L_\Phi}{2}\|x_{k+1} - x_k\|^2$$

$$\leq \Phi(x_k) - \alpha\langle\nabla\Phi(x_k), v_k\rangle + \frac{\alpha^2}{2}L_\Phi\|v_k\|^2$$

$$\leq \Phi(x_k) - \alpha \langle \nabla \Phi(x_k) - v_k, v_k \rangle - \alpha \|v_k\|^2 + \frac{\alpha^2}{2} L_\Phi \|v_k\|^2$$

$$\leq \Phi(x_k) + \frac{\alpha}{2} \|\nabla \Phi(x_k) - v_k\|^2 - \left( \frac{\alpha}{2} - \frac{\alpha^2}{2} L_\Phi \right) \|v_k\|^2$$

$$\leq \Phi(x_k) + \alpha \|\nabla \Phi(x_k) - \widetilde{\nabla} \Phi(x_k)\|^2 + \alpha \|\widetilde{\nabla} \Phi(x_k) - v_k\|^2 - \left( \frac{\alpha}{2} - \frac{\alpha^2}{2} L_\Phi \right) \|v_k\|^2$$

$$\overset{(ii)}{\leq} \Phi(x_k) + \frac{\alpha L'^2}{\mu^2} \|\nabla_y g(x_k, y_k) - \nabla_y g(x_k, y^*(x_k))\|^2 + \alpha \|\widetilde{\nabla} \Phi(x_k) - v_k\|^2$$

$$- \left( \frac{\alpha}{2} - \frac{\alpha^2}{2} L_\Phi \right) \|v_k\|^2$$

$$\overset{(iii)}{\leq} \Phi(x_k) + \frac{\alpha L'^2}{\mu^2} \|\nabla_y g(x_k, y_k)\|^2 + \alpha \|\widetilde{\nabla} \Phi(x_k) - v_k\|^2 - \left( \frac{\alpha}{2} - \frac{\alpha^2}{2} L_\Phi \right) \|v_k\|^2,$$

where $(i)$ follows from Assumptions 2 and 3, $(ii)$ follows from Lemma 7 in [20] and the $\mu$-strong convexity of $g(x, y)$ w.r.t. $y$, and $(iii)$ follows because $\nabla_y g(x_k, y^*(x_k)) = 0$.

Taking the expectation on both sides, we obtain

$$\mathbb{E}[\Phi(x_{k+1})] \leq \mathbb{E}[\Phi(x_k)] + \frac{\alpha L'^2}{\mu^2} \mathbb{E} \|\nabla_y g(x_k, y_k)\|^2 + \alpha \mathbb{E} \|\widetilde{\nabla} \Phi(x_k) - v_k\|^2$$

$$- \left( \frac{\alpha}{2} - \frac{\alpha^2}{2} L_\Phi \right) \mathbb{E} \|v_k\|^2.$$

Then, the proof is complete. $\qquad\qquad\qquad\qquad\qquad\qquad\qquad\qquad\qquad\qquad\qquad \square$

**Lemma 21.** *Suppose Assumptions 1, 2 and 3 hold, $\beta = \frac{2}{13 L_Q}$, $q = (1-a)S_2$, $m = \frac{16}{\mu\beta} - 1$, and $\alpha = \frac{1}{20 L_m^3}$ where $L_m = \max\{L_Q, L_\Phi\}$. Then, we have*

$$\sum_{k=0}^{K-1} \mathbb{E} \|v_k\|^2 \leq L''(\Phi(x_0) - \Phi^*) + \frac{9 L'^2 \alpha \delta_0 L''}{\mu^2} + \frac{18 L'^2 \sigma'^2 K \alpha L''}{\mu^2 S_1}$$

$$+ 2\alpha L'' \sum_{k=0}^{K-1} \|\widetilde{\nabla} \Phi(x_k) - \overline{\nabla} \Phi(x_k)\|^2,$$

*where $\frac{1}{L''} = \frac{\alpha}{2} - \frac{L_\Phi \alpha^2}{2} - \frac{62\alpha^3 L'^2 L_Q^2}{\mu^2} - 44\alpha^3 L_Q^2$, $\sigma'$ is defined in Lemma 18, $\widetilde{\nabla} \Phi(x_k)$ is defined in eq. (11), $\overline{\nabla} \Phi(x_k)$ is defined in eq. (3), and $L'$ is defined in Lemma 18.*

*Proof.* Telescoping eq. (30) over $k$ from 0 to $K - 1$, we have

$$\left( \frac{\alpha}{2} - \frac{L_\Phi \alpha^2}{2} \right) \sum_{k=0}^{K-1} \mathbb{E} \|v_k\|^2$$

$$\leq \Phi(x_0) - \mathbb{E}[\Phi(x_K)] + \frac{\alpha L'^2}{\mu^2} \sum_{k=0}^{K-1} \delta_k + 2\alpha \sum_{k=0}^{K-1} \Delta_k + 2\alpha \sum_{k=0}^{K-1} \|\widetilde{\nabla} \Phi(x_k) - \overline{\nabla} \Phi(x_k)\|^2$$

$$\overset{(i)}{\leq} \Phi(x_0) - \mathbb{E}[\Phi(x_K)] + \frac{\alpha L'^2}{\mu^2} \left( \frac{10\sigma'^2 K}{S_1} + 6\delta_0 + 62\alpha^2 L_Q^2 \sum_{k=0}^{K-2} \mathbb{E} \|v_k\|^2 \right)$$

$$+ 2\alpha \left( \frac{4\sigma'^2 K}{S_1} + 22\alpha^2 L_Q^2 \sum_{k=0}^{K-2} \mathbb{E} \|v_k\|^2 + \frac{4}{3} \delta_0 \right) + 2\alpha \sum_{k=0}^{K-1} \|\widetilde{\nabla} \Phi(x_k) - \overline{\nabla} \Phi(x_k)\|^2$$

$$\leq \Phi(x_0) - \mathbb{E}[\Phi(x_K)] + \left( \frac{10 L'^2}{\mu^2} + 8 \right) \frac{\sigma'^2 K \alpha}{S_1} + \left( \frac{6 L'^2}{\mu^2} + \frac{8}{3} \right) \alpha \delta_0$$

$$+ \left( \frac{62\alpha^3 L'^2 L_Q^2}{\mu^2} + 44\alpha^3 L_Q^2 \right) \sum_{k=0}^{K-2} \mathbb{E} \|v_k\|^2 + 2\alpha \sum_{k=0}^{K-1} \|\widetilde{\nabla} \Phi(x_k) - \overline{\nabla} \Phi(x_k)\|^2,$$

where $(i)$ follow from Lemmas 18 and 19.

We let $\frac{1}{L''} = (\frac{\alpha}{2} - \frac{L_\Phi \alpha^2}{2} - \frac{62\alpha^3 L'^2 L_Q^2}{\mu^2} - 44\alpha^3 L_Q^2)$, which is guaranteed to be positive due to the parameter settings given in the lemma, reorganize the terms in the above inequality, and obtain

$$\frac{1}{L''} \sum_{k=0}^{K-1} \mathbb{E}\|v_k\|^2 \leq \Phi(x_0) - \mathbb{E}[\Phi(x_K)] + \left(\frac{10L'^2}{\mu^2} + 8\right) \frac{\sigma'^2 K\alpha}{S_1}$$

$$+ \left(\frac{6L'^2}{\mu^2} + \frac{8}{3}\right) \alpha\delta_0 + 2\alpha \sum_{k=0}^{K-1} \|\widetilde{\nabla}\Phi(x_K) - \overline{\nabla}\Phi(x_k)\|^2.$$

Then, we have the bound on $\sum_{k=0}^{K-1} \mathbb{E}\|v_k\|^2$ as

$$\sum_{k=0}^{K-1} \mathbb{E}\|v_k\|^2 \leq L''(\Phi(x_0) - \Phi^*) + \frac{9L'^2\alpha\delta_0 L''}{\mu^2} + \frac{18L'^2\sigma'^2 K\alpha L''}{\mu^2 S_1}$$

$$+ 2\alpha L'' \sum_{k=0}^{K-1} \|\widetilde{\nabla}\Phi(x_k) - \overline{\nabla}\Phi(x_k)\|^2.$$

Then, the proof is complete. $\qquad\square$

## D.2   Main Proof of Theorem 2

**Theorem 4.** *(**Formal Statement of Theorem 2***) Apply VRBO to solve the problem in eq.* (1). *Suppose Assumptions 1, 2, 3 hold. Let* $\alpha = \frac{1}{20L_m^3}, \beta = \frac{2}{13L_Q}, S_2 \geq 2(\frac{L}{\mu} + 1)L\beta, m = \frac{16}{\mu\beta} - 1, q = \frac{\mu L\beta S_2}{\mu + L}$, *and* $\eta < \frac{1}{L}$. *Then, we have*

$$\sum_{k=0}^{K-1} \mathbb{E}\|\nabla\Phi(x_k)\|^2 \leq \frac{56L'^2}{\mu^2}\frac{\sigma'^2 K}{S_1} + \frac{30L'^2\delta_0}{\mu^2} + 340\alpha^2 L_Q^2 \frac{L'^2}{\mu^2}\left(L''(\Phi(x_0) - \Phi^*) + \frac{9L'^2\alpha\delta_0 L''}{\mu^2}\right.$$

$$\left. + \frac{18L'^2\sigma'^2 K\alpha L''}{\mu^2 S_1} + 2\alpha L'' C_Q^2 K\right) + 4KC_Q^2,$$

*where* $\frac{1}{L''} = \frac{\alpha}{2} - \frac{L_\Phi \alpha^2}{2} - \frac{62\alpha^3 L'^2 L_Q^2}{\mu^2} - 44\alpha^3 L_Q^2$, $\sigma'$ *is defined in Lemma 18,* $L_m$ *is defined in Lemma 21,* $\widetilde{\nabla}\Phi(x_k)$ *is defined in eq.* (11), $\overline{\nabla}\Phi(x_k)$ *is defined in eq.* (3)*, and* $L'$ *is defined in Lemma 20.*

*Proof.* Based on the form of $\nabla\Phi(x_k)$ in eq. (2), we have

$$\sum_{k=0}^{K-1} \mathbb{E}\|\nabla\Phi(x_k)\|^2$$

$$= \sum_{k=0}^{K-1} \mathbb{E}[\|\nabla\Phi(x_k) - \widetilde{\nabla}\Phi(x_k) + \widetilde{\nabla}\Phi(x_k) - \overline{\nabla}\Phi(x_k) + \overline{\nabla}\Phi(x_k) - v_k + v_k\|^2]$$

$$\leq 4 \sum_{k=0}^{K-1} (\mathbb{E}\|\nabla\Phi(x_k) - \widetilde{\nabla}\Phi(x_k)\|^2 + \mathbb{E}\|\widetilde{\nabla}\Phi(x_k) - \overline{\nabla}\Phi(x_k)\|^2$$

$$+ \mathbb{E}\|\overline{\nabla}\Phi(x_k) - v_k\|^2 + \mathbb{E}\|v_k\|^2)$$

$$\leq 4 \sum_{k=0}^{K-1} (L'^2\|y_k - y^*(x_k)\|^2 + C_Q^2 + \Delta_k^2 + \mathbb{E}\|v_k\|^2)$$

$$\overset{(i)}{\leq} 4 \sum_{k=0}^{K-1} \left(\frac{L'^2}{\mu^2}\|\nabla_y g(x_k, y_k)\|^2 + C_Q^2 + \Delta_k^2 + \mathbb{E}\|v_k\|^2\right)$$

$$\leq 4\sum_{k=0}^{K-1}\left(\frac{L'^2\delta_k}{\mu^2}+C_Q^2+\Delta_k^2+\mathbb{E}\|v_k\|^2\right)$$

$$\overset{(ii)}{\leq}\frac{4L'^2}{\mu^2}\left(\frac{10\sigma'^2K}{S_1}+6\delta_0+62\alpha^2L_Q^2\sum_{k=0}^{K-2}\mathbb{E}\|v_k\|^2\right)+4KC_Q^2+4\sum_{k=0}^{K-1}\mathbb{E}\|v_k\|^2$$

$$+4\left(\frac{4\sigma'^2K}{S_1}+22\alpha^2L_Q^2\sum_{k=0}^{K-2}\mathbb{E}\|v_k\|^2+\frac{4}{3}\delta_0\right)$$

$$\leq\left(\frac{40L'^2}{\mu^2}+16\right)\frac{\sigma'^2K}{S_1}+\left(\frac{24L'^2}{\mu^2}+\frac{16}{3}\right)\delta_0+\left(247\alpha^2L_Q^2\frac{L'^2}{\mu}+88\alpha^2L_Q^2+4\right)\sum_{k=0}^{K-1}\mathbb{E}\|v_k\|^2$$

$$+4KC_Q^2$$

$$\overset{(iii)}{\leq}\frac{56L'^2}{\mu^2}\frac{\sigma'^2K}{S_1}+\frac{30L'^2\delta_0}{\mu^2}+340\alpha^2L_Q^2\frac{L'^2}{\mu^2}\left(L''(\Phi(x_0)-\Phi^*)+\frac{9L'^2\alpha\delta_0L''}{\mu^2}\right.$$

$$\left.+\frac{18L'^2\sigma'^2K\alpha L''}{\mu^2S_1}+2\alpha L''C_Q^2K\right)+4KC_Q^2,$$

where $(i)$ follows from Assumption 1, $(ii)$ follows from Lemma 19, and $(iii)$ follows from Lemma 21. Taking the expectation on both sides, we have

$$\frac{1}{K}\sum_{k=0}^{K-1}\mathbb{E}\|\nabla\Phi(x_k)\|^2\leq\frac{56L'^2}{\mu^2}\frac{\sigma'^2}{S_1}+\frac{30L'^2\delta_0}{\mu^2K}+340\alpha^2L_Q^2\frac{L'^2}{\mu^2K}\left(L''(\Phi(x_0)-\Phi^*)+\frac{9L'^2\alpha\delta_0L''}{\mu^2}\right.$$

$$\left.+\frac{18L'^2\sigma'^2K\alpha L''}{\mu^2S_1}+2\alpha L''C_Q^2K\right)+4C_Q^2.$$

Since $C_Q=\mathcal{O}(1-\eta\mu)^Q, L_Q=\mathcal{O}(Q^2),\beta=\mathcal{O}(Q^{-2}),\sigma'^2=\mathcal{O}(Q^2)$, we obtain the following bound:

$$\frac{1}{K}\sum_{k=0}^{K-1}\mathbb{E}\|\nabla\Phi(x_k)\|^2\leq\mathcal{O}\left(\frac{Q^4}{K}+\frac{Q^6}{S_1}+Q^4(1-\eta\mu)^{2Q}\right).$$

Then, the proof is complete. $\qquad\square$

## D.3 Proof of Corollary 2

**Corollary 4** (**Restatement of Corollary 2**)**.** *Under the same conditions of Theorem 2, choose* $S_1=\mathcal{O}(\epsilon^{-1}), S_2=\mathcal{O}(\epsilon^{-0.5}), Q=\mathcal{O}(\log(\frac{1}{\epsilon^{0.5}})), K=\mathcal{O}(\epsilon^{-1})$. *Then, VRBO finds an $\epsilon$-stationary point with the gradient complexity of $\widetilde{\mathcal{O}}(\epsilon^{-1.5})$ and Hessian-vector complexity of $\widetilde{\mathcal{O}}(\epsilon^{-1.5})$.*

*Proof.* Based on the setting in Corollary 4, we have $\mathcal{O}(\frac{Q^4}{K}+\frac{Q^6}{S_1}+Q^4(1-\eta\mu)^{2Q})=\mathcal{O}(\epsilon)$, which guarantees the target $\epsilon$-accuracy. Note that the period $q=(1-a)S_2=\mathcal{O}(\epsilon^{-0.5})$. Thus, the gradient and Jacobian complexities are given by $\mathcal{O}(KS_1/q+KS_2m)=\widetilde{\mathcal{O}}(\epsilon^{-1.5}+\epsilon^{-1.5})=\widetilde{\mathcal{O}}(\epsilon^{-1.5})$, and that Hessian-vector complexity is given by $\mathcal{O}(KQS_1/q+KS_2mQ)=\widetilde{\mathcal{O}}(\epsilon^{-1.5}+\epsilon^{-1.5})=\widetilde{\mathcal{O}}(\epsilon^{-1.5})$. Then the proof is complete. $\qquad\square$