# OpenReview forum: "Provably Faster Algorithms for Bilevel Optimization"
_NeurIPS.cc/2021/Conference — NeurIPS 2021 Spotlight_

### Official Review · Reviewer_VGzP · 2021-07-08

**Rating:** 7
**Confidence:** 3

**Summary:**

This paper presents two algorithms (one single-loop and one double-loop) which exploit variance reduction techniques to improve the best known sample complexity for stochastic bilevel optimization problems with strongly convex lower-level from $O(\epsilon^{-2})$ to  $O(\epsilon^{-1.5})$.


Bilevel optimization consists in two nested optimization problems: the upper-level aims at minimizing an objective function where some of the variables are a solution of the lower-level problem. In the stochastic setting, upper and lower-level objectives are not accessible and are instead replaced by unbiased estimators. In addition to several smoothness assumptions, this work and several recent others consider the case when the lower-level problem is strongly-convex, which guarantees that the lower-level solution is unique, the existence of the upper-level gradient, and allows to provide convergence guarantees to stationary points even when the lower-level solution is not computed exactly.

This work studies two algorithms (MRBO and VRBO) which both achieve a sample complexity of $O(\epsilon^{-1.5})$, which improves upon the previous optimal complexity of $O(\epsilon^{-2})$. Both algorithms exploit variance reduction  to achieve the desired complexity. MRBO is single loop and combines variance reduction and momentum while VRBO is double loop and uses large-batch gradients to reduce the variance. Recent works have already applied momentum and variance reduction to solve stochastic bilevel problems but this one is the first (together with two concurrent works) to achieve a sample complexity this low. The authors present experiments on the data hyperclearner setting which show that VRBO performs the best and double-loop algorithms seem to outperform single-loop ones.


**Limitations And Societal Impact:**

Unaddressed limitation are at point 3 of the main review.

The authors justify the absence of a discussion about potential negative societal impact writing in the checklist that the nature of the work is theoretical. I agree with the authors.

**Main Review:**

The authors corrected Theorem 1 and 2 with related corollaries in the supplementary. For this reason I will only refer to the restated results of the supplementary.

#### **Originality**
1. Several works have recently tackled the sample complexity of (stochastic) bilevel optimization with smooth objectives and strongly convex lower-level problem. This work, together with two concurrent ones, is the first that achieves a convergence rate of $\tilde O(\epsilon^{1.5})$ using variance reduction techniques. Furthermore, the rate is achieved  with two algorithms (MRBO and VRBO). MRBO is very similar to the concurrent work SUSTAIN [20], while VRBO is the first that combines double-loop and variance reduction.  The contributions are stated clearly and related work is properly cited.

#### **Quality**
2. The theoretical claims (Corrected in the appendix) seem more or less correct, although I checked just a little part of the proofs also because they were not easy to follow (see major comment 8) and I am not so familiar with variance reduction techniques.
3.  Drawbacks of the analysis could be discussed more. For example the fact that some parameters like step sizes or batch sizes depend on unknown constants and on the accuracy threshold. Also, I think that adding an additional discussion of the assumptions will be valuable since they seem quite restrictive.
4. Experimental evaluation is narrow. Only one setting (data hypercleaner) is presented, despite that the strength of  the bilevel framework is that it can be applied to several practical scenarios. The authors could either comment on this in the contributions section or perform experiments on at least another setting like optimizing the regularization hyperparameter. On the positive side, they compare their methods with several others from recent literature.

#### **Clarity**
5. The main ideas of the paper are easy to grasp for someone acquainted with recent works in bilevel convergence rates. The paper contains numerous typos (see minor comments) and the proofs are quite hard to follow (see major comment 8). The work could also benefit from simple examples of bilevel optimization problems that could make it more accessible to people less familiar with the bilevel setting.

#### **Significance**
6. Bilevel optimization is becoming more and more relevant in machine learning since many problems like instances of meta-learning and hyperparameter optimization can be cast into this framework. This work develops two principled and efficient algorithms that improve the optimal rate for stochastic bilevel optimization with a strongly convex lower level problem. Although the experimental evaluation is narrow, the results are promising and show the potential advantage of these approaches in practice.

#### **Final Score**
7. I am giving 6 to the paper but I could increase the score if the authors improve the proofs section in the supplementary, comment more on the drawbacks of the analysis, or improve the experimental section.


#### **Major comments**
8. Proofs (in the supplementary) are quite hard to follow, they contain some typos and several references to lemmas contained in recent related works. I think they need more polishing and at least some referenced lemmas could be written in the supplementary to make it more self-contained.
9. I think the complexity reported in the abstract and in the contribution should be up to logarithmic terms ($\tilde O$ instead of $O$) to account for the hessian-vector products, which generally are more expensive than gradients.
10. This work and related ones lack a more detailed discussion on the assumptions which it looks like they can be too restrictive in some cases unless the optimization problem(s) is over a convex compact set, a setting which this work does not cover.
11. Since the authors mention concurrent work, it will bring value to the paper to include it not only in lines 131-142 but throughout the manuscript when comparing with related works, for example in Table 1.
12. I think that the Justification in lines 100-105 is not well supported experimentally. Furthermore, it should not be in the contribution section.
13. The concurrent methods SUSTAIN [20] and BSA [8] can be also applied with bigger batch sizes, while the authors seem to only consider batch size = 1 for these methods in the experiments (from the provided code). I think it will be good to see the performance of these methods with larger batch sizes. Probably in this case SUSTAIN will be equivalent to MRBO.


#### **Minor Comments  and typos**
14. From Theorem 2 statement it is unclear why the final complexity in Corollary 2 $O(\epsilon^{1.5})$ holds, probably because the dependence on $\mathcal{S}_2$ and $q$ is hidden in (12). It seems that the complexity is just $\tilde{O}(\epsilon^{-1})$. I do not know if there is a simple way to make it more clear.
15. Experiments could be better detailed in the appendix by specifying what metric is used to fine-tune the parameters for best performance and what kind of CPU is used.
16. I think adding some plots with the number of samples on the x axis to have results which are independent of the particular implementation and hardware can add value to the paper.
17. Contribution section could be shorter by removing some details.
18. Line 203. I think Assumption 1 should not include $\Phi$ being non-convex. The results would hold also for convex $\Phi$.
19. Assumption 2 requires everything to be uniform also w.r.t. the random variables. This may be commented on because it is not present in works which instead do not use variance reduction.
20. VRBO (Algorithm 2) could be written more clearly since it seems that uses different $\tilde x$-s inside the $t$ for loop but then writes $\tilde x_{k,t+1} = \tilde x_{k,t}$. It would be clearer if $\tilde x $ had only the $k$ subscript.
21. I would change “by the order of magnitude” in line 72.
22. Line 87 “periodically at each outer loop” is unclear.
23. Line 92: unclear what “Hessian-vector type” means.
24. Line 100: “outer-loop estimation”?
25. Line 101: “inner loop output y” not clear, i would remove “y”.
26. Line 109: “to further simplify the implementation”?
27. order-wisely -> order-wise at lines 115, 174, 350.
28. equations (2) and (3) could be with $x$ instead of $x_k$. Furthermore in (3) the last product could probably be replaced by an exponent since the index $j$ is not used.
29. Line 231 “of dataset” can be removed
30. Line 239: “one step iteration” -> “one iteration”
31. whether -> when at line 68.

#### **Post Rebuttal**
The authors have properly addressed my concerns and claimed to make several (quite straightforward) modifications for the next version of the paper. Thus I am increasing the score from 6 to 7.


#### **References**
[8] S. Ghadimi and M. Wang. Approximation methods for bilevel programming.arXiv preprint376arXiv:1802.02246, 2018.

[12] Z. Guo and T. Yang. Randomized stochastic variance-reduced methods for stochastic bilevel386optimization.arXiv preprint arXiv:2105.02266, 2021.

[19] P. Khanduri, S. Zeng, M. Hong, H.-T. Wai, Z. Wang, and Z. Yang. A momentum-assisted single-timescale stochastic approximation algorithm for bilevel optimization. arXiv preprint arXiv:2102.07367v1, 2021.

[20] Khanduri, P., Zeng, S., Hong, M., Wai, H.-T., Wang, Z., and Yang, Z. (2021). A near-optimal algorithm for stochastic bilevel optimization via double-momentum.arXiv preprint arXiv:2102.07367


**Time Spent Reviewing:**

12

---

> ### Author Response · Authors · 2021-08-10
> **Response Part II (Response to Minor Comments and typos)**
>
> Q14. From Theorem 2 statement it is unclear why the final complexity in Corollary 2 $O(\epsilon^{-1.5})$ holds, probably because the dependence on $S_2$ and $q$ is hidden in (12). It seems that the complexity is just $\tilde{O}(\epsilon^{-1})$. I do not know if there is a simple way to make it more clear.
>
> A: Sorry about the confusion. The statement of Corollary 2 in the main text contains typos, and the correct version is given in Corollary 4 (a restatement of Corollary 2 in the supplementary) in lines 772-781. As stated in Corollary 4, the choices of $S_1$, $S_2$, $Q$ and $K$ should be $O(\epsilon^{-1})$, $O(\epsilon^{-0.5})$, $O(\log (\epsilon^{-0.5}))$ and $O(\epsilon^{-1})$. Then $q$ has the order of $O(\epsilon^{-0.5})$ due to its dependence on $S_2$, and $m$ has the order of $O(1)$ due to its dependence on $Q$. Then the complexity $\tilde{O}(\epsilon^{-1.5})$ follows. We will revise these accordingly.
>
> Q15. Experiments could be better detailed in the appendix by specifying what metric is used to fine-tune the parameters for best performance and what kind of CPU is used.
>
> A: Many thanks for this good suggestion! In our experiments, we used the training loss as the metric to tune the parameters for best performance. All experiments are performed on an iMac with 3.8GHz Quad-Core Intel Core i5 CPU and 32 GB 2400 MHz DDR4. We will add these details in the revision.
>
> Q16. I think adding some plots with the number of samples on the x axis to have results which are independent of the particular implementation and hardware can add value to the paper.
>
> A: Many thanks for this suggestion. We will add experiments using the number of samples as the x axis.
>
> Q17. Contribution section could be shorter by removing some details.
>
> A: Many thanks for this suggestion! We will shorten the Contribution section by removing some algorithm and technical details, and will move the justification in lines 100-105 to the experiment section as suggested in Q12.
>
> Q18. Line 203. I think Assumption 1 should not include $\Phi$ being non-convex. The results would hold also for convex $\Phi$.
>
> A: Many thanks for this suggestion! We will remove the assumption on nonconvexity of $\Phi$.
>
> Q19. Assumption 2 requires everything to be uniform also w.r.t. the random variables. This may be commented on because it is not present in works which instead do not use variance reduction.
>
> A: Many thanks for pointing this out! We will clarify the requirement of Assumption 2 in the revision, and will discuss the difference of this assumption from those in other studies. Interestingly, we find that this assumption has also been adopted by some works that did not use variance reduction such as stocBiO [17], BSA [8], STABLE [2], MSTSA [19], SEMA [11].  (References are from the reference list of the paper.)
>
> Q20-21: Suggestions about math expressions and wording
>
> A: Many thanks for the suggestions! We will revise our paper accordingly.
>
> Q22. Line 87 “periodically at each outer loop” is unclear.
>
> A: Many thanks for pointing this out. “Periodically at each outer loop” here means that after every fixed number (e.g., $q$ in our context) of iterations, we compute a large-batch hypergradient estimation to reduce the variance. We will clarify this in the revision.
>
> Q23. Line 92: unclear what “Hessian-vector type” means.
>
> A: Many thanks for pointing this out. “Hessian-vector type” means that our constructed hypergradient estimator in (3) involves only the computation of Hessian-vector products rather than Hessians. The computation procedure is detailed in Appendix A. We will clarify this in the revision.
>
> Q24. Line 100: “outer-loop estimation”?
>
> A: Many thanks! “Outer-loop estimation” means the hypergradient estimation at the outer loop. We will clarify this in the revision.
>
> Q25. Line 101: “inner loop output y” not clear, I would remove “y”.
>
> A:  Thanks and we will revise our paper accordingly.
>
> Q26. Line 109: “to further simplify the implementation”?
>
> A: Thanks and we will revise “to further simplify the implementation” to “to further simplify the implementation of constraint-based bilevel methods”.
>
> Q27-31: Suggestions on wording and math expressions.
>
> A: Many thanks for these good suggestions! We will make the changes accordingly.

---

> ### Author Response · Authors · 2021-08-10
> **Response Part I (Response to Major Comments)**
>
> Many thanks for the expert review!
>
> Q3. Drawbacks of the analysis could be discussed more. For example the fact that some parameters like step sizes or batch sizes depend on unknown constants and on the accuracy threshold. Also, I think that adding an additional discussion of the assumptions will be valuable since they seem quite restrictive.
>
> A: Many thanks for the suggestion! We will add a paragraph to discuss the analysis drawbacks and how to address them potentially. For example, it will be interesting to exploit parameter-free approaches (e.g., via line search or using history information) to remove the unknown constants in our stepsizes and batchsizes.
>
> We will also add a section to discuss the assumptions as follows. Assumptions 2 and 3 essentially require the Lipschitzness of gradients and second-order derivatives of inner and outer objective functions. Assumption 2 also requires the gradient of the outer objective function to be bounded. These assumptions have been adopted by all existing studies for stochastic bilevel optimization. Further, these assumptions are mild in practice as long as the iterates along the practical training path are bounded. For example, all our experiments indicate that these iterates are well located in a bounded regime. Another way to relax the boundedness assumption is to consider a bilevel problem with a convex compact set.  Our analysis for the unconstrained setting in this paper can be extended to study such a constrained case by introducing a projection into the iterative updates.
>
> Q4. Experimental evaluation is narrow. Only one setting (data hypercleaner) is presented, despite that the strength of the bilevel framework is that it can be applied to several practical scenarios. The authors could either comment on this in the contributions section or perform experiments on at least another setting like optimizing the regularization hyperparameter. On the positive side, they compare their methods with several others from recent literature.
>
> A: We thank the reviewer for pointing this out. We are currently working on implementing the experiment of optimizing the regularization hyperparameters on NewsGroup dataset. The results (coming out so far) agree with the experiment presented in the paper. Namely, our VRBO converges initially slower than stocBiO but faster later on in the high accuracy regime, and VRBO achieves higher accuracy on test data than stocBiO (VRBO 61% v.s. stocBiO 57%). The experimental comparison with other algorithms is still ongoing.
>
> Q5. The main ideas of the paper are easy to grasp for someone acquainted with recent works in bilevel convergence rates. The paper contains numerous typos (see minor comments) and the proofs are quite hard to follow (see major comment 8). The work could also benefit from simple examples of bilevel optimization problems that could make it more accessible to people less familiar with the bilevel setting.
>
> A: We thank the reviewer for all suggestions on our presentation. We will carefully proofread our paper and correct the typos and reorganize our proof structure to make it more readable. We will include the hyper-cleaning application (one of our experiments) as an illustration example to help readers better understand the bilevel setting.
>
> Q8. Proofs (in the supplementary) are quite hard to follow, they contain some typos and several references to lemmas contained in recent related works. I think they need more polishing and at least some referenced lemmas could be written in the supplementary to make it more self-contained.
>
> A: Thanks for this good suggestion! We will add all referenced lemmas in the supplementary material to make the proof self-contained, and will carefully reorganize our proof structure to make it easier to follow. We will also carefully proofread all proofs and correct typos.
>
> Q9. I think the complexity reported in the abstract and in the contribution should be up to logarithmic terms ($\tilde{O}$ instead of $O$) to account for the hessian-vector products, which generally are more expensive than gradients.
>
> A: Many thanks for this good suggestion! We fully agree and will revise our paper accordingly.
>
> Q10.  This work and related ones lack a more detailed discussion on the assumptions which it looks like they can be too restrictive in some cases unless the optimization problem(s) is over a convex compact set, a setting which this work does not cover.
>
> A: Many thanks for this good point! We will add a section to discuss all assumptions. Please see our response to Q3 for more details about our discussion of the assumptions.
>
> Q11. Since the authors mention concurrent work, it will bring value to the paper to include it not only in lines 131-142 but throughout the manuscript when comparing with related works, for example in Table 1.
>
> A: Many thanks for the suggestion!  We will include all concurrent works’ results in our comparison throughout the paper including Table 1.
>
> Q12. I think that the Justification in lines 100-105 is not well supported experimentally. Furthermore, it should not be in the contribution section.
>
> A: Many thanks for this suggestion! The justification in lines 100-105 serves as a possible reason for the observations in the experiments, which we will clarify in the revision. As suggested, we will remove it from the contribution section.
>
> Q13. The concurrent methods SUSTAIN [20] and BSA [8] can be also applied with bigger batch sizes, while the authors seem to only consider batch size = 1 for these methods in the experiments (from the provided code). I think it will be good to see the performance of these methods with larger batch sizes. Probably in this case SUSTAIN will be equivalent to MRBO.
>
> A: Many thanks for this suggestion! In experiments, we agree that SUSTAIN with general batchsize will be equivalent to MRBO, and hence will have the same experimental performance. We will clarify this in the revision. We currently choose the batch size to be 1 for SUSTAIN and BSA because these algorithms and the corresponding theorems were developed in the single-sample version.

---

> ### Comment · Reviewer_VGzP · 2021-08-19
> **Updated Review and Score**
>
> Thank you very much for the comprehensive response. I updated the review increasing the score.

---

> > ### Author Response · Authors · 2021-08-25
> > **Many thanks for your further updates!**
> >
> > We thank the reviewer very much for further reviewing our response and raising the score!

---

### Official Review · Reviewer_5eE2 · 2021-07-14

**Rating:** 7
**Confidence:** 3

**Summary:**

The paper extends to usual algorithms of convex optimization with finite sum to bilevel optimization: SGD with momentum and SVRG. The main contribution of the paper theoretical: authors derive accelerated convergence rates toward the solution of the bilevel problem.


**Main Review:**


Theoretical results are interesting for the bilevel community, although this work does not fix the practical problem of hand-tuned stepsizes.

Comments:
- In algorithm 1 and 2, how do you choose the stepsize s $\lambda$, $\gamma$, $\alpha$ and $\beta$ in practice. Do you have any insights on how to set in practice? From my experience, algorithms are very sensitive this hyperparameters.
- How do you select $Q$ in practice? (the size of the "Neumann serie")
From my experience it can also have a large influence of the result of the bilevel optimization algorithm.
How robust is the algorithm to this parameter?
- Since computing $\hat \nabla \Phi(x_k, B_k)$ (through (4)) requires an iterative procedure, I think it is misleading to talk about "single" loop algorithm for Algorithm 1, and "double loops" algorithm for Algorithm 2, do authors agree that the computation of $\hat \nabla \Phi(x_k, B_k)$ add another loop?
- Authors claim that the proposed techniques are order of magnitude faster than usual algorithms, regarding the experimental results, I would be less assertive.


**Time Spent Reviewing:**

8h

---

> ### Author Response · Authors · 2021-08-10
> **Many thanks for your expert review**
>
> Many thanks for the expert review!
>
> Q: In algorithm 1 and 2, how do you choose the stepsizes $\lambda$, $\gamma$, $\alpha$ and $\beta$ in practice. Do you have any insights on how to set in practice? From my experience, algorithms are very sensitive to these hyperparameters.
>
> A: Good question! In our experiments, for our Algorithms 1 and 2, we apply the standard grid search when the stepsizes $\lambda$, $\gamma$, $\alpha$ and $\beta$ are all chosen from the interval [1e-3,1], and select those that yield the best convergence performance.
>
> In practice, we find that an algorithm with lower variance is more robust (i.e., less sensitive) to the choice of the stepsize. In our problem, we find that the performance of our algorithms is more robust to the stepsize selection than STABLE, because our algorithms apply Hessian-vector estimation which has much smaller variance than the explicit Hessian computation used in STABLE. Specifically, our algorithms MRBO and VRBO converge stably when $\alpha$ and $\gamma$ are chosen from a large interval $[0.01,1]$, and their best performance do not change much when $\alpha$ and $\gamma$ are chosen from $[0.1,0.3]$, whereas STABLE can converge stably only when the stepsize is as small as $\alpha$ = 1e-10.
>
> Q: How do you select $Q$ in practice (the size of the "Neumann serie")? From my experience it can also have a large influence on the result of the bilevel optimization algorithm. How robust is the algorithm to this parameter?
>
> A: Good question! In practice, we tend to choose a relatively smaller $Q$ to attain a fast convergence speed. For example, we choose $Q$ from {1,2,3} in our experiments following all benchmark algorithms such as AID-FP, reverse, and stocBiO. We have also tried larger $Q$ (e.g., 5,10,15,20) in our experiments, where we find that the final test accuracy remains the same but the convergence is dramatically slow. Therefore, our algorithms are robust to $Q$ in terms of the test accuracy, but a smaller $Q$ chosen from 1 to 5 is preferred in practice to attain fast convergence.
>
> Q: Since computing $\hat{\nabla}\Phi(x_k;B_k)$ (through (4)) requires an iterative procedure, I think it is misleading to talk about "single" loop algorithm for Algorithm 1, and "double loops" algorithm for Algorithm 2, do authors agree that the computation of $\hat{\nabla}\Phi(x_k;B_k)$ add another loop?
>
> A: We fully agree with this good point! Indeed, considering the computation of $\hat{\nabla}\Phi(x_k;B_k)$, which involves another loop with size of $Q$, MRBO and VRBO should be regarded as double- and triple-loop algorithms. We will clarify this in our revision. (In this rebuttal, to avoid confusion, we will still use the same notion of loops as the paper.)
>
> Q: Authors claim that the proposed techniques are order of magnitude faster than usual algorithms, regarding the experimental results, I would be less assertive.
>
> A: Very good point! In fact, the theoretical order-level improvement tends to be more observable in experiments in the high accuracy regime (when the target accuracy $\epsilon$ becomes small). Thus, in our experiments (see Figure 1), our proposed VRBO performs much better (i.e., attains a much lower training loss) than SGD-type stocBiO in a relatively high-accuracy regime (i.e., after 10s in Figure 1(a)), whereas it is a little bit slower than stocBiO during the initial training period. Such a phenomenon is also commonly observed in the conventional minimization and minimax optimization problems. This also suggests an interesting problem for a future study: exploring a hybrid method by combining the initial fast training of stocBiO and the accurate convergence of VRBO.
>
> We also find that the number of loops can also significantly affect the practical performance of the algorithms. Double-loop algorithms tend to outperform single-loop algorithms of the same type. Thus, although our double-loop VRBO outperforms all algorithms, our single-loop MRBO converges slightly slower than double-loop algorithms such as AID-FP and stocBiO. We suspect that this is because double-loop designs are more suitable for bilevel optimization than single-loop schemes, as we comment in the last paragraph of Section 1.1. This happens to all recently designed momentum-based single-loop algorithms such as MSTSA, STABLE, and SUSTAIN. In fact, among all these single-loop algorithms, our MRBO performs the best.

---

> > ### Comment · Reviewer_5eE2 · 2021-08-25
> > **Comments on author response**
> >
> > Comment:
> >
> > I thank the authors for their response, and I will raise my score. I think it is important to highlight these numerical details: the double and triple loop, the choice of the "outer gradient step", and the choice of Q.
> >
> > In particular it seems to me that there is a dichotomy in the bilevel community: proof are often based of choice of Q large enough, but in order to be efficient in practice, Q is always chosen small.
> > I would like to know if authors agree with my last remark.
> > If yes, I think it would be good for the community to highlight it in the paper.

---

> > > ### Author Response · Authors · 2021-08-25
> > > **Many thanks for your further comments!**
> > >
> > > We thank the reviewer very much for the further comments and raising the score!
> > >
> > > Q: I think it is important to highlight these numerical details: the double and triple loop, the choice of the "outer gradient step", and the choice of Q.
> > >
> > > A: Thanks for this great suggestion! In the revision, we will highlight the numerical details as pointed out by the reviewer. We will specify that MRBO has double loops, and VRBO has triple loops. We will further elaborate all these loops. We will also specify the choice of the stepsize for the outer gradient step and the choice of Q, in both the theorems and the experiments.
> > >
> > > Q: In particular it seems to me that there is a dichotomy in the bilevel community: proof are often based of choice of Q large enough, but in order to be efficient in practice, Q is always chosen small. I would like to know if authors agree with my last remark. If yes, I think it would be good for the community to highlight it in the paper.
> > >
> > > A: Many thanks for pointing out the dichotomy of Q for bilevel optimization problems! We fully agree with this remark and will also highlight it in the revised paper. This comment also inspires us to study in the future whether it is possible to establish the theoretical convergence for smaller Q.

---

### Official Review · Reviewer_YFpX · 2021-07-19

**Rating:** 6
**Confidence:** 3

**Summary:**

The authors developed two algorithms MRBO and VRBO for  bilevel stochastic optimization problem, and showed that their computational complexities ($\mathcal{O}(\epsilon^{-1.5})$) outperform all existing algorithms ($\mathcal{O}(\epsilon^{-2})$) orderwisely. The authors claim two main contributions. First, MRBO is the first momentum algorithm that exhibits the orderwise improvement over SGD-type algorithms for bilevel optimization. Second, VRBO is the first that adopts the recursive variance reduction technique to accelerate bilevel optimization. Their experiments demonstrate that these algorithms  outperform existing algorithms, and suggest that the double-loop design may be more suitable for bilevel optimization than the single loop structure.

**Limitations And Societal Impact:**

Although the proposed algorithms are demonstrated in simple MNIST tasks, more experiments on large scale data (e.g., CIFAR or ImageNet) are necessary to further valid the computational efficiency.

**Main Review:**

Originality:
The authors proposed two new algorithms for solving stochastic bilevel optimization program: MRBO is the first momentum algorithm that exhibits the orderwise improvement over SGD-type algorithms for bilevel optimization; VRBO is the first that adopts the recursive variance reduction technique to accelerate bilevel optimization. Both of these two algorithms improve the computational efficiency by a order of $\mathcal{O}(\epsilon^{0.5})$.


Quality:
Both algorithms are supported with proofs of convergence guarantees. Also, experiments on simple MNIST tasks are conducted to demonstrate the superior performance of the proposed algorithms.

Clarity:
This paper is well-organized and  provides clear statements of algorithms and proofs.

Significance:
Bilevel optimization is an important tool for many ML tasks such as hyperparameter tunning. However, it is also well-know solving Bilevel optimization program is NP-hard. This paper proposes two algorithms that are well suited for a class of such programs.

**Time Spent Reviewing:**

2 hours

---

> ### Author Response · Authors · 2021-08-10
> **Many thanks for your expert review**
>
> Many thanks for the expert review!
>
> Q: Although the proposed algorithms are demonstrated in simple MNIST tasks, more experiments on large scale data (e.g., CIFAR or ImageNet) are necessary to further validate the computational efficiency.
>
> A: Thanks for this good suggestion! We will run more experiments on CIFAR or ImageNet datasets over Convolutional Neural Networks, and will include the experimental results in the revision.
>
> For your information, we are currently working on implementing the experiment of optimizing the regularization hyperparameters on NewsGroup dataset. The results (coming out so far) agree with the experiment presented in the paper. Namely, our VRBO converges initially slower than stocBiO but faster later on in the high accuracy regime, and VRBO achieves higher accuracy on test data than stocBiO (VRBO 61% v.s. stocBiO 57%). The experimental comparison with other algorithms is still ongoing.

---

### Decision · Program_Chairs · 2021-09-27

**Decision:**

Accept (Spotlight)

**Comment:**

Reviewers unanimously agree that this is a paper of high originality and soundness of the results. Some reviewers have raised concerns over inflated claims in light of experimental validation (5eE2, VGzP), as well clarity of the proofs and discussion of the limitations (VGzP). After a strong rebuttal and engaging in discussion with the authors, the reviewers have found that the authors have addressed their remark, and some reviewers increased their score.